# Stable long-term individual differences in 50-kHz vocalization rate and call subtype prevalence in adult male rats: Comparisons with sucrose preference

**Adithi Sundarakrishnan** [ORCID]**, Paul B. S. Clarke** [ORCID] *

Department of Pharmacology and Therapeutics, McGill University, Montréal, Québec, Canada

* paul.clarke@mcgill.ca

**Data Availability Statement:** Our dataset of acoustic (WAV) files is very large (800+ files, totalling approx. 400 GB). We will share these WAV files upon request. The numerical data derived

## Abstract

Sucrose preference (SP) is a widely used measure of anhedonia in rat models of depression, yet depressed patients do not reliably show an analogous deficit. As an alternative affect-related measure, adult rat ultrasonic vocalizations (USVs) are attracting interest, but it is unclear whether SP and USVs provide independent measures. Here, we have assessed whether SP and USV emission are correlated in the absence of a depressogenic procedure. To this end, 24 male Long-Evans rats were tested daily for 24 days, with alternating SP tests and USV recordings; after a 3-month hiatus, USV emission was re-evaluated for 6 more days. SP was measured in simultaneous two-bottle choice tests, and USVs were recorded in an open field. The main measures were: SP, 50-kHz call rate, and relative prevalence of trill and flat call subtypes. These measures showed temporally-stable individual differences across the initial 24-day testing period, and at the 3-month USV follow-up tests. Correlational analysis revealed no significant relationships between SP and the three main USV measures. Rats differed consistently, not only in their 50-kHz call rates but also in their 50-kHz call profiles (i.e., the relative prevalence of 14 call subtypes); most rats preferentially emitted either trill or flat calls. Several inter-call subtype associations were detected, including a strong negative relationship between the relative prevalence of flat and trill calls. The 50-kHz call rate was correlated with the relative prevalence of only one call subtype (short calls, negative correlation), but was positively correlated with absolute emission rates for almost all subtypes. In conclusion, adult rats exhibited temporally-stable individual differences over weeks (SP) or months (USVs) of testing. This trait-like stability helped to reveal a lack of relationship between SP and the USV-related variables under study, suggesting that these measures may capture different constructs of possible relevance to animal models of depression.

## Introduction

Adult rat ultrasonic vocalizations (USVs) are attracting interest as potential markers of negative and positive affect [1]. It was initially proposed that 22-kHz calls reflect negative affect [2],

from these WAV files, together with the raw data from SP tests, can be found at 10.6084/m9.figshare.20341206.

**Funding:** Funded by a Discovery Grant (15505, awarded to PBSC) from the Natural Sciences and Engineering Research Council of Canada (NSERC, https://www.nserc-crsng.gc.ca/). The funders had no role in study design, data collection and analysis, decision to publish, or preparation of the manuscript.

**Competing interests:** The authors have declared that no competing interests exist.

whereas 50-kHz calls [3]—and more specifically frequency-modulated 50-kHz calls [4]—captured positive affect. However, these proposals represent an overgeneralization, and in particular fail to consider the rich heterogeneity of 50-kHz calls. Notably, among the 14 or more call subtypes identified to date [5–8], only two have been consistently associated with positive *vs.* negative affect: trill and flat calls, respectively. Specifically, the trill call subtype is reported to predominate when rats were given euphorigenic drugs (amphetamine, cocaine, chronic morphine [5, 9, 10]) or the opportunity to socially interact [5, 11], whereas the trill calls were suppressed by an aversive dose of the opioid antagonist naloxone when given alone [12]. Conversely, the flat call subtype became more prevalent during acute morphine withdrawal [12] and after administration of dopamine D1 receptor antagonists [13].

USV-derived measures show evidence of long-term "trait-like" stability in adult rats. Temporally-stable individual differences have been reported mostly in terms of 50-kHz call rates [14–21], with evidence largely provided by studies in which low *vs.* high rates of 50-kHz call emission (defined by a median split) were either found to be associated with an independent behavioural measure [20, 22], or else served as a guide for selective breeding [23, 24]. In such studies, 50-kHz USV data were collapsed across the time frame of the experiment. In contrast, we have previously explored inter-rat variability across several USV recording sessions, reporting clear temporally-stable individual differences for 50-kHz call rate and also for acoustic features such as frequency bandwidth [5]. In a preliminary analysis, we also suggested that rats may differ consistently in their 50-kHz *call profile*, i.e., the relative prevalence of different call subtypes [5].

To date, almost all investigations of individual differences in USV emission have concerned stimulus-evoked 50-kHz calling. Exceptionally, Schwarting et al. (2007) showed short-term individual differences in rates of 50-kHz calling recorded in home cages as well as novel cages, across four daily tests [14]. Therefore, to extend these findings, the present study focused on USV emission recorded over *several weeks and months* under relatively neutral conditions, with an emphasis on call subtype analysis.

Negative affect, a key construct in major depressive disorder, is particularly challenging to model in rodents. Notably, the commonly used sucrose preference (SP) test is regarded as a measure of anhedonia, i.e., an impaired ability to derive pleasure from rewarding stimuli [25]. However, depressed patients do not reliably show an analogous deficit [26–29]. Several published reports have investigated USV emission within a depression model in adult rats [19, 20, 30–33]. In all studies, USVs were experimentally evoked—either by acute administration of amphetamine [30, 31], tickling [19, 33] anticipation or experience of play [20], or by the presence of sucrose [32].

The relationship between individual differences in SP *vs.* USV emission has not been thoroughly investigated, with the focus largely confined to 50-kHz call *rate*. Most relevant studies have included a depressogenic procedure, and whereas some have revealed a positive association between SP and 50-kHz call rate [24, 32], others have not [19, 30, 31]. The power of this correlative approach depends critically on the extent to which individual differences in SP reflect a stable trait rather than random day-to-day fluctuation. As with USVs (see above), some temporal stability in SP is implied by studies using the median split approach [30, 34–38], but there appear to be few published studies in which SP was reported across multiple days in *individual animals* [39, 40]. Finally, one study has reported both SP and emission rates for individual 50-kHz call subtypes [31]. Here, chronic variable stress reduced SP and suppressed 50-kHz call emission, but with no clear call subtype-selective effect.

In the present study, we investigated whether SP and 50-kHz USV measures possess long-term trait-like properties under relatively neutral conditions, in the absence of any drug or depressogenic procedure. We also asked whether individual differences in SP were related to

individual differences in USV measures. Based on prior findings (see above), three USV measures were of primary interest: the 50-kHz call rate, and the relative prevalence of flat and trill calls. However, analysis was extended to include other 50-kHz call subtypes. The three primary USV measures were found to be uncorrelated to SP, suggesting that one or more may potentially serve as an independent measure of affect in longer-term experiments.

## Materials and methods

### Subjects

Twenty-four adult male Long-Evans rats (Charles River Laboratories, Kingston, NY, USA) were used, housed two per cage in a humidity- and temperature-controlled colony room. Rats were left to acclimate to the colony room for 7 days upon arrival, and were subsequently handled by the experimenter for 4 days. Handling sessions consisted of 5–7 minute of interaction with each pair of rats. At the start of Phase 1, rats weighed 270–314 g, corresponding to approximately 8 weeks of age (based on data from supplier). At the start of Phase 2, rats weighed 503–719 g. Home cage and test cage bedding consisted of laboratory-grade Beta Chip® (NEPCO, Warrensburg, NY). Rats were maintained on a reverse 12:12-h light/dark cycle with the lights off at 07h00, and all testing was performed between 10h00 and 18h00 of the dark phase of the cycle. Food and water were available *ad libitum* except during test sessions, or where noted. At the end of testing, subjects were euthanized with $CO_2$ exposure. All procedures were approved by the McGill Animal Care Committee in accordance with the guidelines of the Canadian Council on Animal Care.

### Experimental design

In overview, the study comprised two Phases (Fig 1). In Phase 1, each rat was given four habituation sessions and was then tested daily for a total of 24 days. Phase 2 started after an interval of 13 weeks, and comprised brief re-habituation followed by six USV recording sessions.

In more detail, Phase 1 started with four once-daily sessions which served to habituate subjects first to the SP test apparatus (two sessions) and then to the USV apparatus (two sessions). Over the next 24 days, rats received one session each day, with SP tests alternating with USV recording sessions (Fig 1). On a given day, all the animals received either a SP test session or a USV recording session. Hence, in Phase 1 there were a total of 12 SP tests and 12 USV recording sessions. Phase 1 revealed marked and stable inter-individual differences in 50-kHz call rates and call profiles, across the 24 days of testing, and so in Phase 2, we determined whether these USV-related individual differences would be conserved over a longer time period. Accordingly, USV recordings were resumed after an interval of 13 weeks, using the same testing procedure as before. Phase 2 comprised two initial re-habituation sessions (2 days)

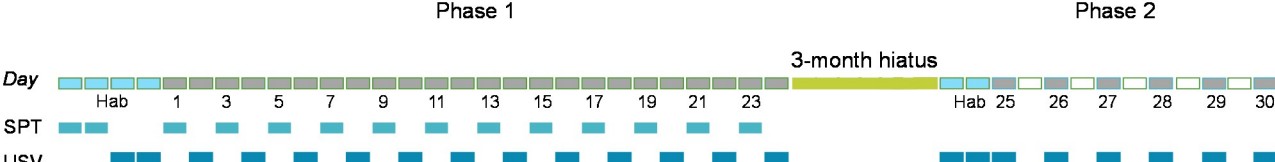

**Fig 1. Timeline of habituation and testing.** Rats were tested for a total of 30 sessions (i.e., 12 SP and 18 USV). Phase 1 comprised four habituation sessions followed by 24 days of once-daily testing, with sucrose preference (SP) tests and USV recording sessions interleaved. Phase 2 began 3 months after the end of Phase 1 and lasted 11 days; this phase comprised two habituation sessions followed by six USV recording sessions (grey boxes) occurring at two-day intervals.

followed by six once-daily sessions occurring at 2-day intervals. During Phase 2, SP was not tested.

## Sucrose preference (SP) test

Sucrose preference tests were conducted in the colony room between 10h00 and 18h00, during the dark phase of the cycle. Rats were tested individually in semi-transparent polypropylene cages (48.3 cm long, 26.7 cm wide, and 20.3 cm high; Ancare R20 Rat cage). The cages were fitted with a stainless-steel wire lid (Ancare R20 wire lid) in which the two bottles were laid next to each other. The bottle spouts were separated by approximately 6 cm. All habituation sessions were of 120 min duration and test sessions were of 60 min duration.

During the two SP habituation sessions, rats were given simultaneous access to two bottles of water (Day 1) or two bottles of 0.3% sucrose solution (Day 2). During subsequent test sessions, a two-bottle simultaneous choice test was conducted to measure preference for 0.3% sucrose solution *vs*. tap water. The solutions of 0.3% (w/v) sucrose (Millipore Sigma; CAS number 57-50-1) were prepared fresh each day, in tap water. Before the start of the test session, bottles were weighed and left at room temperature for 30 min.

Daily SP tests were performed in two sessions, with 12 rats per session. A given rat was always tested in the same daily session and in the same cage. Before the start of a test session, the 12 rats were transferred to the 12 test cages and given 30 minutes to habituate, without access to any fluids. Next, the two bottles were placed in the test cage, and during the 1-hour test session, the animals had free access to both bottles. The position of the bottles was counterbalanced across test sessions to account for possible side preferences. Fluid consumption was calculated as the difference in bottle weight before *vs*. after the session. After the test session, rats were returned to their home cages, and one hour later were again given *ad libitum* access to food and water, to avoid an association between return of food and the SP test. At the end of each test day, half of the test cage bedding was replaced with fresh bedding, in order to retain a relatively constant bedding condition across days.

Fluid spillage due to bottle handling was estimated prior to the start of the experiment and averaged 0.96 mL. All sucrose and water intake values were corrected for spillage, before the daily sucrose preference was calculated.

## Ultrasonic vocalizations: Apparatus, recording and analysis

Ultrasonic vocalization recordings were conducted in a testing room between 12h00 and 18h00. All habituation and test sessions were of 20 minutes duration. Before a session, subjects were transported from the colony room in their home cages, and allowed to settle for 20–90 min in a room adjacent to the testing room, with ad-libitum access to food and water. Rats were returned to the colony room after the session.

At the start of a USV recording session, each rat was placed individually in one of four test boxes. A given rat was always tested in the same box. These boxes were rectangular (58 cm long, 29 cm wide, and 53 cm high) and comprised four vertical white-painted walls made of melamine-coated fiberboard, and an 8-mm-thick clear Plexiglas™ lid, attached by a hinge. The four test boxes were placed in a quadrant configuration, approximately 2.5 cm apart, on a layer of Beta Chip® bedding. At the end of each test day, half of the bedding was replaced with fresh bedding, in order to retain a relatively constant bedding condition across days.

Broadband recordings and acoustic analysis were performed as detailed in our recent publications (e.g., Best et al. 2017). An ultrasonic condenser microphone (CM16/CMPA, Avisoft Bioacoustics, Berlin, Germany), was placed in an opening (9 cm x 6 cm) cut halfway along one long side of the lid and connected to an UltraSoundGate 416H data acquisition device (Avisoft

Bioacoustics). To check microphone sensitivity, a custom-made ultrasound emitter was used that generated trains of 50-kHz pulses of variable intensity. This device confirmed that 50-kHz calls were detectable from all locations within a given test box, but not registered by microphones in adjacent test boxes. The sampling rate was 250-kHz with 16-bit resolution. Spectrograms were generated by fast Fourier transform (512 points, 75% overlap, FlatTop window, 100% frame size) using Avisoft SASLab Pro (v. 5.2.09). Ultrasonic calls were manually selected from spectrograms (by A.S.), and no call intensity threshold was applied. 22-kHz calls (i.e., 20–25 kHz) were treated as a single type, whereas 50-kHz calls (i.e., 25–95 kHz) were categorized according to our previously-published 14-subtype scheme [5]. After initial inspection, it was evident that some rats made hundreds of 50-kHz calls per session, and hence it was decided to time-sample recordings from high-calling rats. More specifically, we time-sampled all sessions (i.e., in Phases 1 and 2), from any rat that emitted more than 100 calls per 20-min session on the final two USV sessions. Based on this criterion, 11 out of the 24 rats were time-sampled. In these rats, selection and subtyping was restricted to calls whose onset fell within the middle minute of every 3-min period, i.e., 60–120, 240–300, 420–480, 600–660, 780–840, 960–1020, and 1140–1200 s.

## Statistical analysis

Data were analyzed using Systat software (v. 11, SPSS, Chicago, IL) and Prism 9 (GraphPad Software, La Jolla, CA). Figures were generated using Prism 9 (GraphPad Software, La Jolla, CA). Primary measures of interest were percent sucrose preference, the 50-kHz call rate, and the percentages of flat and trill calls. To minimize any existing side preference, SP values were averaged across two consecutive days to form day pairs. In order to facilitate comparison with SP, USV-related measures were also broken down by day pair. 50-kHz call profiles, defined by the proportional contributions of all 14 call subtypes [5], were calculated for each rat separately, by pooling calls within day pairs and giving each call (rather than each session) equal weight. Hence, the *relative* prevalence of a given 50-kHz call subtype on a given day pair was calculated as the number of calls of that subtype, divided by the total number of 50-kHz calls, expressed as a percentage. As a rate-free measure, relative (i.e. percent) prevalence is not skewed by high-calling rats. For comparison, we also calculated the *absolute* prevalence of each 50-kHz call subtype, expressed as the number of calls/min.

In SP tests, 3 out of 288 sessions yielded extremely high values of total fluid intake (i.e., 27.5, 42.6 and 45.3 mL per 60-min session). These values, which occurred in rats showing otherwise moderate intake, were many times higher than the group median, and represent approximately 24 hours of normal fluid consumption in non-deprived rats of the same strain and age [41, 42]; on this basis, they most likely represent fluid leakage rather than true consumption, and were therefore excluded from the statistical analyses. For 50-kHz call profiles, calculation was rendered less reliable in a few test sessions by low numbers of 50-kHz calls. Specifically, 5 out of the 24 rats emitted fewer than 5 calls in at least one day pair; however, these low-calling occurrences were relatively few, i.e., 6/144 and 4/72 sessions in Phases 1 and 2, respectively. These low-calling sessions were retained in all analyses, since removing them did not affect the results significantly.

Since parametric test assumptions were often violated, nonparametric tests were used throughout. In order to test for individual (i.e. inter-rat) differences, Kruskal-Wallis tests were used, with one of our four primary measures (e.g. SP) serving as the dependent variable, and RAT as the between-subject factor. To further assess the temporal stability of individual differences, Cronbach's alpha values were calculated; here, the dependent variable consisted of a given primary measure collected across successive day pairs, with rat serving as the

experimental unit, as before. To compare performance on USV measures between Phases 1 and 2, Wilcoxon matched pairs tests were used. Spearman's correlational analysis was used to evaluate potential relationships between measures (for details, see Results section). To assess Spearman correlations from Phase 1 and Phase 2 together, Fisher combined p-values were generated. An alpha level of 1% (2-tailed) was chosen for all statistical tests.

## Results

### Main USV measures (50-kHz call rate, prevalence of flat and trill calls)

In order to assess shorter-term stability of individual differences, data were first analyzed separately for Phases 1 and 2. Longer-term stability was then assessed by comparing USV-derived measures between Phases 1 and 2.

**Phase 1.** In Phase 1, the total numbers of 22-kHz and 50-kHz calls identified were 85 and 28,456, respectively. 22-kHz calls were sparse, representing only 0.3% of all calls, and were emitted sporadically by all 24 rats.

Performance on each of the three main USV measures was maintained across the 6 day pairs during the 24-day testing period (Fig 2). Pronounced and temporally stable individual differences were seen with respect to 50-kHz call rate (Fig 3A, S1A, S1B, S2A and S2B Figs) and to the percentage prevalence of flat and trill calls (Fig 3C and 3E, S1C–S1F and S2C–S2F Figs). These individual differences were statistically significant for all three measures, based on Kruskal-Wallis tests: 50-kHz call rate (H(24) = 105.1, p < 0.0001), percent flat calls (H(24) = 79.6, p < 0.0001), and percent trill calls (H(24) = 90.8, p < 0.0001). Corresponding Cronbach's α values were all high: 0.91, 0.83 and 0.89.

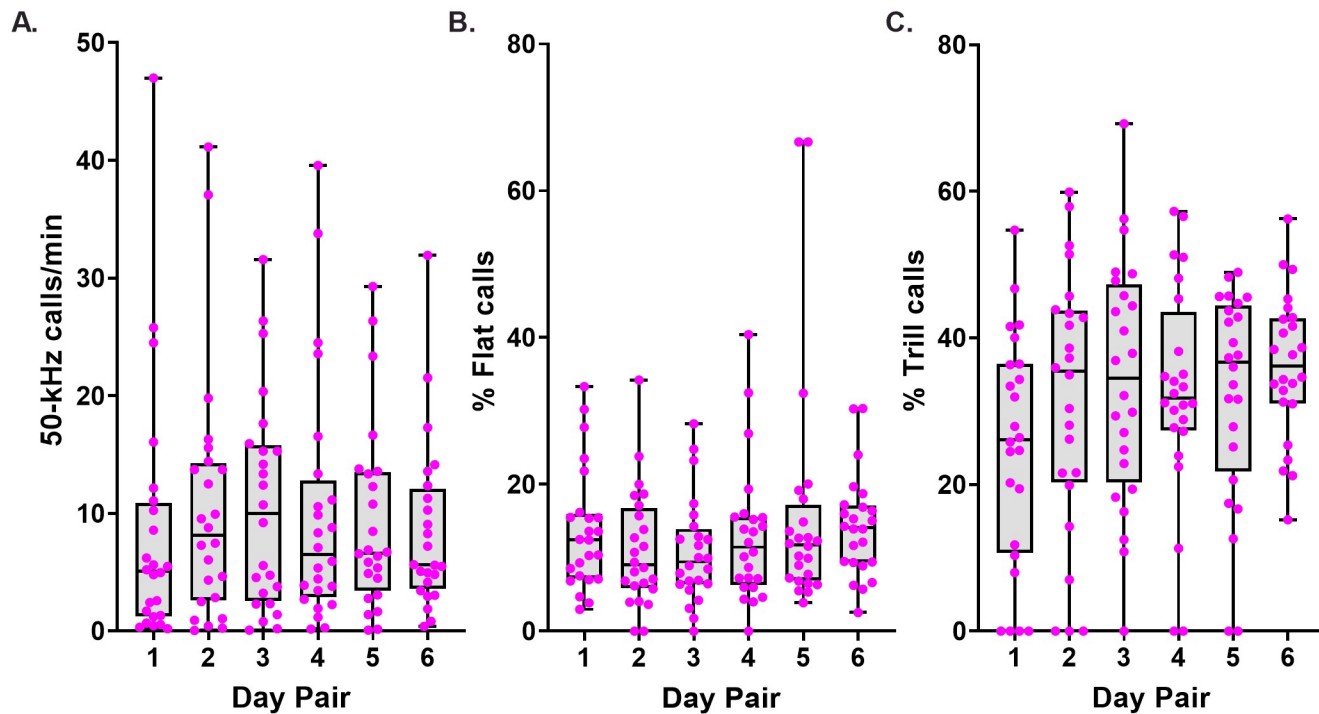

**Fig 2. Phase 1 USV measures.** Phase 1 USV measures: 50-kHz call rate, percentage of flat calls and trill calls across day pairs. Box plots show median, lower and upper quartiles, and minimum and maximum values (n = 24 rats). (A) Calls per minute. (B and C) Percentage prevalence of flat calls and trill calls, respectively.

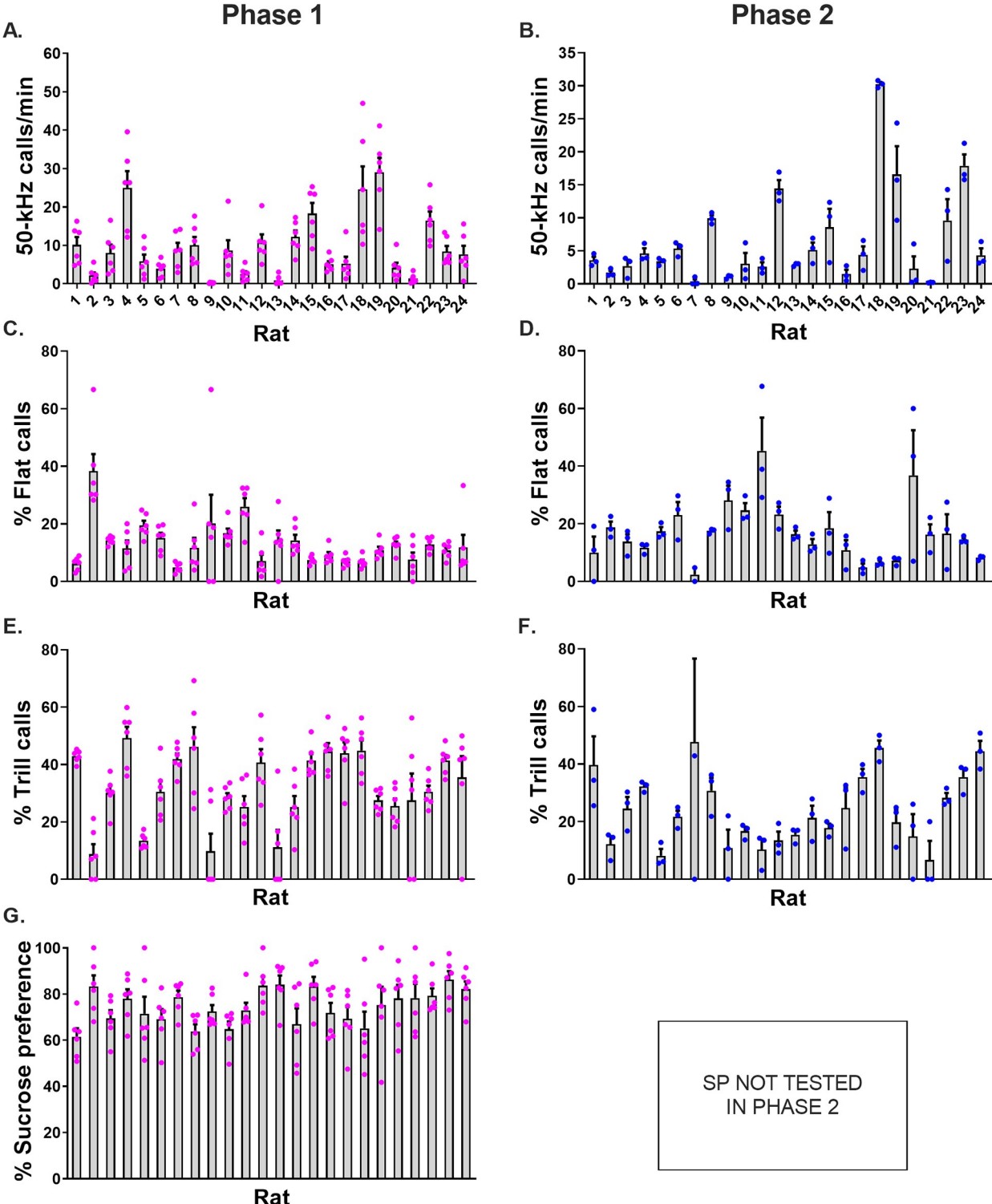

**Fig 3. Individual differences in sucrose preference (SP) and ultrasonic vocalization (USV) measures.** In Phase 1, both SP and USVs were recorded whereas in Phase 2, only USVs were recorded. All panels show data of individual rats. For each behavioural measure, pairs of days were averaged, and each day pair is represented by a coloured symbol. Bars show mean + SEM (n = 6 day pairs per rat). (A and B) 50-kHz call rates in Phase 1 and 2, respectively. (C and D) Percentage prevalence of flat calls in Phase 1 and 2, respectively. (E and F) Percentage prevalence of trill calls in Phase 1 and 2, respectively. In panel F, one high value (rat 7, day pair 3, 100% prevalence) was omitted from the figure, but included in the statistical analysis. (G) Sucrose preference in Phase 1.

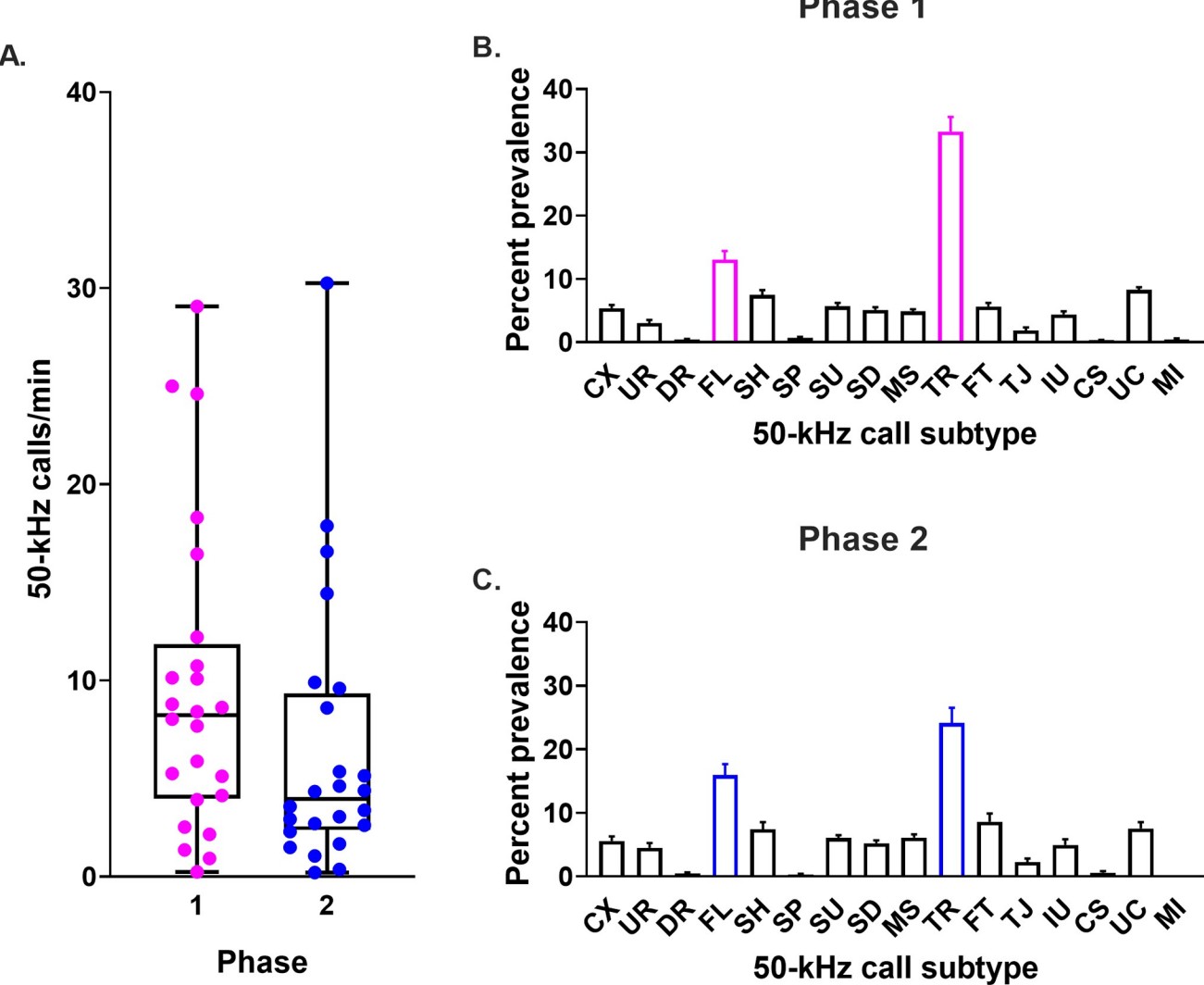

**Fig 4. Group-averaged USV measures.** Group-averaged USV measures: 50-kHz call rate and 50-kHz call profile in Phases 1 and 2. (A) Box plot showing 50-kHz call rate (median, lower and upper quartiles, and minimum and maximum values). (B and C) The y-axes show the mean + SEM prevalence of each 50-kHz call subtype in Phase 1 and 2, respectively. Call subtype definitions are as follows; CX: complex, UR: upward ramp, DR: downward ramp, FL: flat, SH: short, SP: split, SU: step-up, SD: step-down, MS: multi-step, TR: trill, FT: flat-trill, TJ: trill with jumps, IU: inverted-U, CS: composite, UC: unclear, MI: miscellaneous. The group-averaged 50-kHz call subtype profiles were similar in Phases 1 and 2, except that trill calls were less prevalent in the latter. n = 24 rats.

To assess whether these individual differences were already evident early in the testing period, USV performance on each individual day pair was compared with the corresponding performance averaged across the remaining 5 day pairs. Statistically significant correlations were found for all day pairs with respect to all three main USV measures: 50-kHz call rate ($r_s$ = 0.59 to 0.89, n = 24 rats, p = 0.0025 to p < 0.0001), flat calls ($r_s$ = 0.52 to 0.70, n = 24 rats, p = 0.0098 to p = 0.0002) and percent trill calls ($r_s$ = 0.52 to 0.76, n = 24 rats, p = 0.0091 to p < 0.0001).

Overall, the most prevalent 50-kHz call subtypes were flat and trill calls (Fig 4B), with trills predominating over flat calls (group means: 33% *vs.* 13%). The percentages of flat and trill calls were not significantly correlated with the 50-kHz call rate ($r_s$ = -0.44, p = 0.0314 and $r_s$ = 0.47, p = 0.0193, flats and trills respectively, n = 24 rats; Fig 5A and 5C).

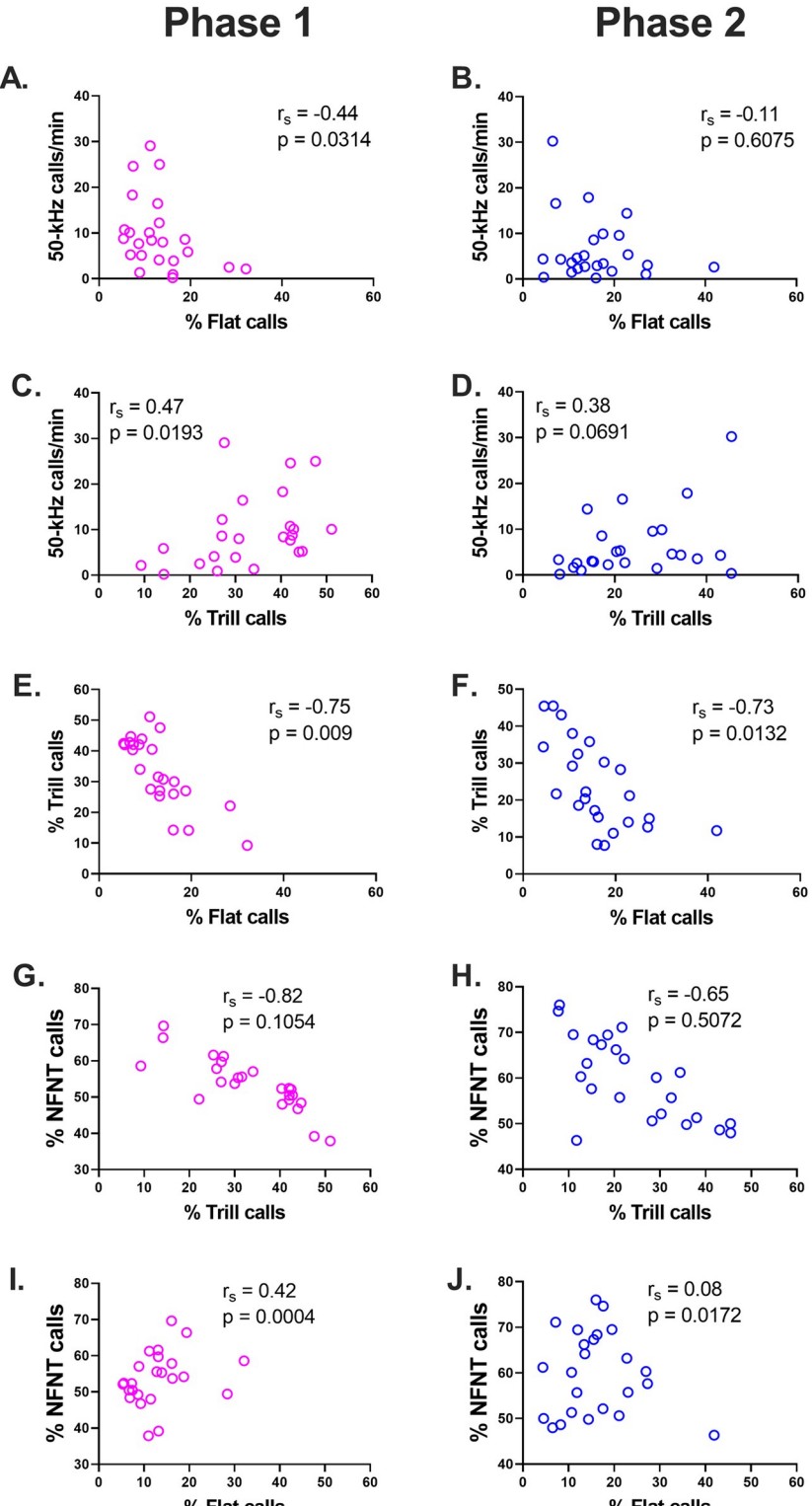

**Fig 5. The relationships between 50-kHz call rate, and prevalence of flat, trill and other calls.** Symbols represent individual rats. (A and B) show the 50-kHz call rate vs. percentages of flat calls in Phases 1 and 2, respectively. (C and D) show the corresponding relationship for trill calls. Within each phase, these call subtype measures were not significantly correlated with 50-kHz call rate (rs, Spearman correlation coefficient). (E to J) show p values based on simulations, as explained in the main text (Results, section called Comparisons between all 50-kHz call subtypes) and

detailed in S2 Table. (E and F) show an inverse relationship between the relative prevalence of trill vs. flat calls; this relationship was statistically significant. (G and H) In contrast, the apparent inverse relationship between grouped non-flat/non-trill (i.e., NFNT) calls and trill calls was not statistically significant, based on simulations. (I and J) show a positive correlation between non-flat/non-trill (i.e., NFNT) calls and flat calls. n = 24 rats in all panels.

**Phase 2.** In the 6 days of testing in Phase 2, the total number of 22-kHz and 50-kHz calls identified was 217 and 10,419, respectively. Similar to Phase 1, 22-kHz calls were emitted by a majority of the rats (19/24 rats), but were again sparse, representing only 2% of all calls. The 50-kHz call rate was lower in Phase 2 compared to Phase 1 (Fig 4A). As in Phase 1, fewer flat calls were emitted than trill calls (16% *vs.* 24%, respectively; Fig 4C); however, the group mean percentage of trill calls was less than in Phase 1 (Fig 4B *vs.* 4C). In Phase 2, the prevalence of flat or trill calls was once again not significantly correlated with the 50-kHz call rate ($r_s$ = -0.11, p = 0.6075 and $r_s$ = 0.38, p = 0.0691, respectively, n = 24 rats; Fig 5B and 5D).

Pronounced individual differences were seen for all three main USV-related measures (Fig 3B, 3D and 3F): 50-kHz call rate (Kruskal-Wallis: H(24) = 58.36, p < 0.0001), percent flat calls (H(24) = 47.22, p = 0.0021) and percent trill calls (H(24) = 49.26, p = 0.0011). These findings were also reflected in the Cronbach's α values for call rate, flat and trill calls, respectively (i.e., 0.95, 0.72 and 0.85). S3 and S4 Figs show the performance of individual rats across successive day pairs.

**Phase 1 *vs.* 2.** Compared to Phase 1, rats tested in Phase 2 emitted a lower percentage of trill calls (Wilcoxon matched pairs test: Z = 7.628, p < 0.0001, Fig 6C), while also tending to emit fewer 50-kHz calls and a higher percentage of flat calls (Z = 2.171, p = 0.0229 and Z = 0.8332, p = 0.0894, respectively; Fig 6A and 6B).

Individual differences in the three main USV measures were, to a large extent, preserved between the two testing phases. Thus, USV emission was significantly correlated between Phases 1 and 2 with respect to call rate ($r_s$ = 0.71, n = 24 rats, p < 0.0001, Fig 6A), percent flat calls ($r_s$ = 0.64, n = 24 rats, p = 0.0008, Fig 6B) and percent trill calls ($r_s$ = 0.76, n = 24 rats, p < 0.0001, Fig 6C). The corresponding Cronbach's α values were moderately high (0.79, 0.78 and 0.83, respectively).

The subtype profiles of each rat in Phase 1 and Phase 2 are shown in Fig 7 and S5 Fig, respectively. Most rats preferentially emitted the trill subtype in both phases (19/24 rats and

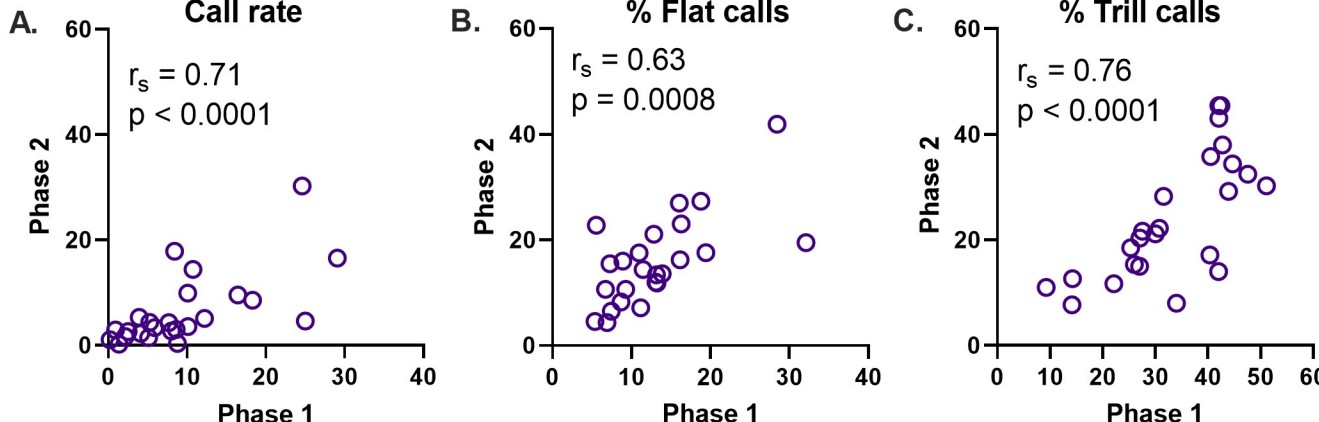

**Fig 6. Stable long-term individual differences in 50-kHz USV measures (Phase 1 vs. Phase 2).** The three main USV measures are shown: 50-kHz call rate (A), percentage prevalence of flat calls (B) and percentage prevalence of trill calls (C). Each rat is represented by a single line. $r_s$, Spearman correlation coefficient, n = 24 rats.

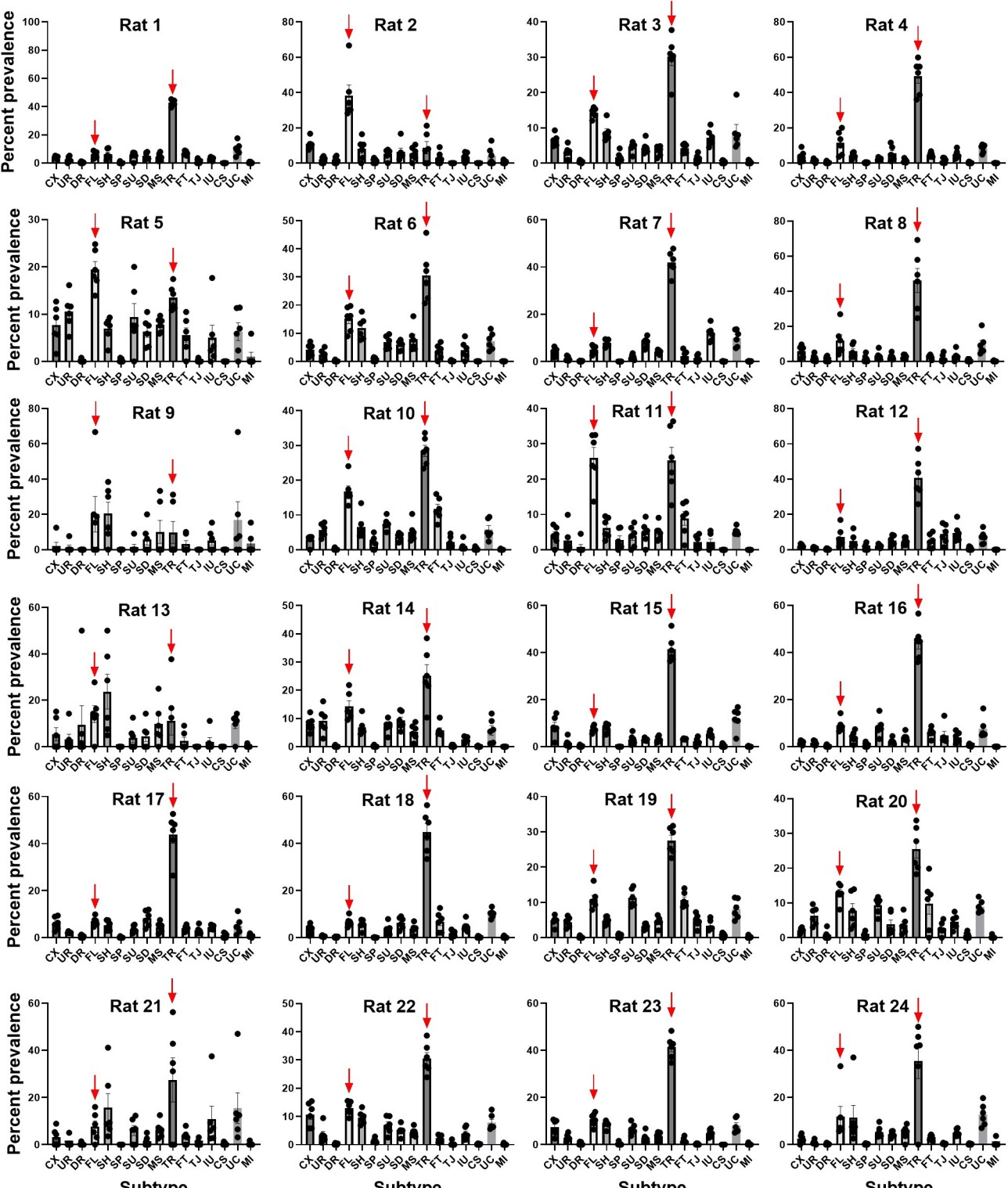

**Fig 7. 50-kHz call subtype profiles of individual rats in Phase 1.** The mean + SEM percentage prevalence of each call subtype is shown (n = 6 day pairs). Most rats showed a preference for trill calls over flat calls. Red arrows indicate flat and trill calls. Call subtype definitions are as follows; CX: complex, UR: upward ramp, DR: downward ramp, FL: flat, SH: short, SP: split, SU: step-up, SD: step-down, MS: multi-step, TR: trill, FT: flat-trill, TJ: trill with jumps, IU: inverted-U, CS: composite, UC: unclear, MI: miscellaneous.

13/24 rats, respectively). Flat calls were the second most preferred subtype (3/24 rats and 9/24 rats, respectively). Exceptionally, two rats preferentially emitted another subtype: short calls in one case (Fig 7, Rat 13, S5 Fig, Rat 9) and inverted-U calls in the other (Rat 21, phase 2, shown in S5 Fig).

### Sucrose preference

In the six pairs of counterbalanced SP tests that comprised Phase 1, temporally consistent individual differences were observed (Kruskal-Wallis: H (24) = 46.81, p = 0.0024; Fig 3G). This temporal stability was reflected by a moderate Cronbach's α value of 0.61. When SP on each day pair was compared with SP averaged across the other 5 day pairs, the first day pair 1 showed the lowest correlation coefficient (day pairs 1–6, respectively: $r_s$ = 0.24, 0.39, 0.33, 0.61, 0.37, and 0.36, n = 24 rats). S6 Fig shows the performance of individual rats across successive day pairs.

### Lack of relationship between sucrose preference and USV measures

Between-rat variability was larger for the three USV measures than for SP (Fig 3A, 3C, 3E *vs.* 3G). This difference was reflected in the larger coefficients of variation seen for the USV variables call rate, percent flat calls and percent trill calls (i.e., 0.83, 0.51 and 0.34, respectively) compared to the SP variable (i.e., 0.10).

Sucrose preference was not significantly correlated with any of the three main 50-kHz USV measures, i.e., call rate, percent flat and trill calls (respectively: $r_s$ = -0.13, -0.08 and -0.18, n = 24 rats, p = 0.5327, 0.7253 and 0.3977; Fig 8A–8C). In an exploratory analysis, we found no significant relationships between SP and the percentage prevalence of any other, less common, 50-kHz call subtypes ($r_s$ = -0.35 to 0.39, n = 24 rats, p > 0.05 for all; S1 Table).

Sucrose *intake*, a secondary measure, appeared to be associated with sucrose preference ($r_s$ = 0.51, n = 24 rats, p = 0.0118, S7A Fig), but not with the main USV measures, i.e., call rate, percent flat and percent trill calls (respectively: $r_s$ = -0.27, -0.06 and -0.10, n = 24 rats, p > 0.1 for all; S7B–S7D Fig).

### Comparisons between all 50-kHz call subtypes

Two kinds of correlational analysis were used to examine relationships between 50-kHz call subtypes. In the first approach, we employed simulations based on random numbers drawn

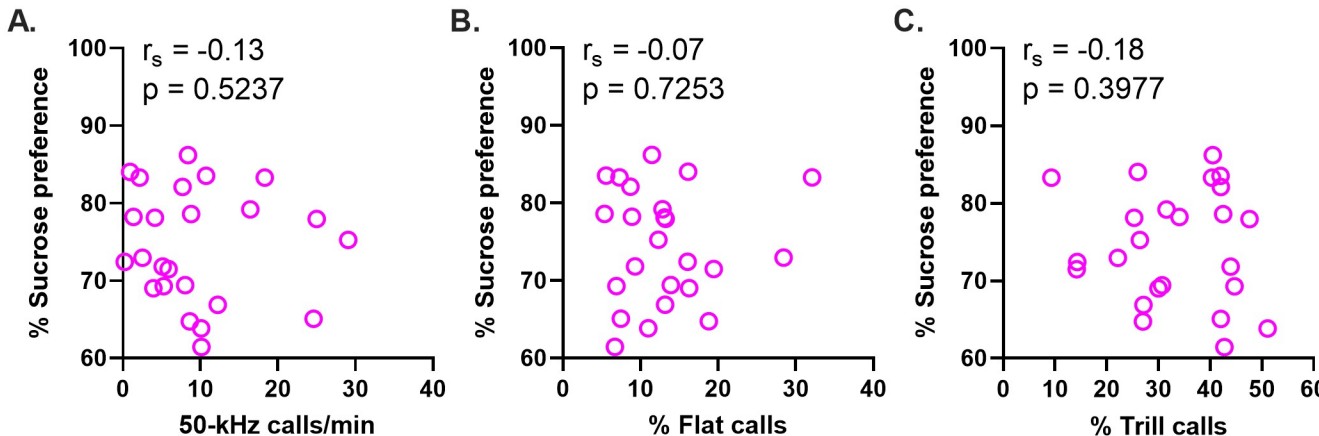

**Fig 8. Individual differences in sucrose preference vs. the three main USV measures.** There was no detectable relationship between sucrose preference and the 50-kHz call rate (A), percentage of flat calls (B), or percentage of trill calls (C) ($r_s$, Spearman correlation coefficient, n = 24 rats).

from Gaussian distributions based on our observed mean ± SD percentage prevalence values. To this end, we first simplified the 50-kHz call profile by reducing it to three categories, i.e. flat, trill and a third category comprising all remaining call subtypes. This third category was termed "NFNT" (i.e. non-flat/non-trill). All three call categories appeared normally distributed in both Phases 1 and 2 (Kolmogorov-Smirnov tests, p > 0.25 in all six cases). In Phase 1, the observed percentage prevalences were as follows: 13.0 ± 6.67, 33.3 ± 11.3 and 53.7 ± 7.4 (mean ± SD for flat, trill and NFNT calls, respectively). The corresponding values in Phase 2 were 16.0 ± 8.5, 24.1 ± 11.8 and 59.9 ± 9.1.

This simulation-based approach was first applied to our Phase 1 data (Table 1). In our observed data set, trill calls were negatively correlated with both flat calls ($r_s$ = -0.75; Fig 5E) and NFNT calls ($r_s$ = -0.82; Fig 5G). In contrast, flat calls were positively correlated with NFNT calls ($r_s$ = +0.42; Fig 5I). We next asked how often such extreme values would occur through random variation alone. To this end, we simulated 5,000 experiments, each with 24 subjects, using random numbers generated from three Gaussian distributions, each representing a different call category and based on the corresponding mean ± SD values noted above. These simulations showed that Spearman $r_s$ values based on random numbers demonstrated a strong negative bias: trill *vs*. flat $r_s$ = -0.39 ± 0.18, trill *vs*. NFNT $r_s$ = -0.68 ± 0.13, flat *vs*. NFNT $r_s$ = -0.29 ± 0.20 (mean ± SD, n = 5000 iterations). We next assessed statistical significance by determining the proportion of simulated $r_s$ values that exceeded our observed values. This analysis showed that the negative correlation observed between trill and flat calls was significantly more negative than expected by chance ($r_s$ = -0.75, p = 0.0090), whereas the observed positive correlation between flat and NFNT calls was significantly more positive than expected by chance ($r_s$ = 0.42, p = 0.0004). In contrast, the strong negative correlation observed between trill and NFNT calls ($r_s$ = -0.82) was not statistically significant (p = 0.1054).

Broadly similar results were obtained in Phase 2 (Table 1, Fig 5 right column). In this second Phase, simulated $r_s$ values were (mean ± SD, 5000 iterations): -0.37 ± 0.19 (trill *vs*. flat), -0.64 ± 0.14 (trill *vs*. NFNT) and -0.37 ± 0.19 (flat *vs*. NFNT). Taking the two Phases together,

**Table 1. Spearman rho ($r_s$) correlation coefficients and Fisher combined p-values for inter-call subtype relationships.**

|  | Subtype name | Rho value Phase 1 | Rho value Phase 2 | p-value Phase 1 | p-value Phase 2 | Fisher combined p-value* |
|---|---|---|---|---|---|---|
| **Approach 1** |  |  |  |  |  |  |
| Trill *vs*. | flat | -0.7530 | -0.7330 | 0.0090 | 0.0132 | 0.0012 |
| Trill *vs*. | non-trill/non-flat | -0.8235 | -0.6496 | 0.1054 | 0.5072 | 0.2100 |
| Flat *vs*. | non-trill/non-flat | 0.4191 | 0.0817 | 0.0004 | 0.0172 | 0.0001 |
| **Approach 2** |  |  |  |  |  |  |
| Flat *vs*. | upward ramp | 0.5930 | 0.3096 | 0.0023 | 0.1410 | 0.0029 |
| Flat *vs*. | split | 0.3528 | 0.6786 | 0.0908 | 0.0003 | 0.0003 |
| Flat *vs*. | trill with jumps | -0.5735 | -0.5223 | 0.0034 | 0.0088 | 0.0003 |
| Flat *vs*. | inverted-U | -0.5887 | -0.4626 | 0.0025 | 0.0228 | 0.0006 |
| Flat *vs*. | composite | -0.5116 | -0.1482 | 0.0106 | 0.4895 | 0.0325 |
| Flat *vs*. | unclear | -0.4678 | -0.5452 | 0.0212 | 0.0059 | 0.0012 |
| Trill *vs*. | upward ramp | -0.4983 | -0.2165 | 0.0132 | 0.3096 | 0.0266 |
| Trill *vs*. | trill with jumps | 0.5231 | 0.2358 | 0.0087 | 0.2673 | 0.0164 |
| Trill *vs*. | inverted-U | 0.5487 | 0.4087 | 0.0055 | 0.0474 | 0.0024 |
| Trill *vs*. | unclear | 0.5939 | 0.4983 | 0.0022 | 0.0132 | 0.0003 |

*For approach 2, this table shows only the call subtype comparisons that yielded statistically significant (p < 0.01) or trending (0.01 < p < 0.05) Fisher combined p-values for Phases 1 and 2.

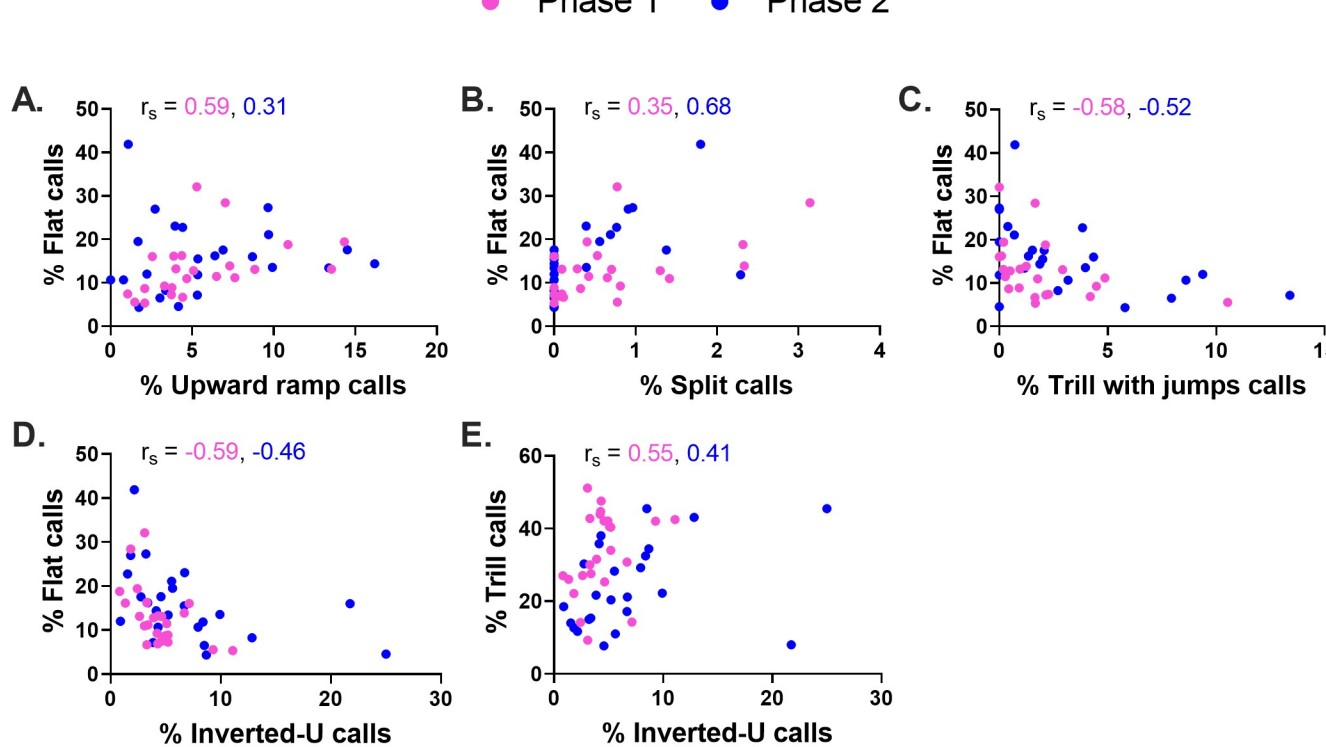

**Fig 9. Statistically significant inter-call subtype correlations in Phase 1 and Phase 2.** The percentage of flat calls and trill calls were significantly correlated with the percentage prevalence of several subtypes (now expressed as a percentage of all non-flat/non-trill 50-kHz calls), in both Phases. Flat calls were related to: upward ramp calls (A), split calls (B), trill with jumps calls (C), inverted-U calls (D). Trill calls were related to: inverted-U calls (E) ($r_s$, Spearman correlation coefficient, n = 24 rats).

this simulation-based approach confirmed that the negative relationship between trill and flat calls was stronger than expected through chance alone.

In the second correlational approach, we asked whether a high prevalence of flat or trill calls was associated with a disproportionate shift in the prevalence of any other 50-kHz call subtypes. To this end, we determined Spearman correlation ($r_s$) coefficients relating flat or trill call prevalence (expressed, as before, as a percentage of *all* 50-kHz calls) to the prevalence of every other 50-kHz call subtype (now expressed as a percentage of all *non-flat/non-trill* 50-kHz calls). This analysis, which was performed separately for Phases 1 and 2, produced several significant ($p<0.01$) correlations and trends ($0.01 < p < 0.05$), as seen in Table 1, S2 Table and Fig 9. When the two Phases were considered together using Fisher combined p-values (Table 1), the percentage prevalence of flat calls was found to be positively correlated with upward ramp ($p = 0.0029$) and split ($p = 0.0003$) calls, and was negatively correlated with trill with jumps ($p = 0.0003$), inverted-U ($p = 0.0006$), and calls in the "unclear" category ($p = 0.0012$). The percentage prevalence of trill calls, in turn, was positively correlated with inverted-U ($p = 0.0024$) and unclear ($p = 0.0003$) categories; there was also a positively trending association between trills and trill with jumps ($p = 0.0164$). Lastly, there was a notable absence of correlation between flat-trill combination calls and either flat or trill subtypes (S2 Table).

## 50-kHz call rate *vs.* absolute and relative call subtype prevalence

Finally, two exploratory correlational analyses were conducted in order to look for possible relationships between the 50-kHz call rate and the prevalences of individual 50-kHz call

**Table 2. Spearman correlation coefficients relating 50-kHz call rate to the relative or absolute prevalence of individual 50-kHz call subtypes.**

| | Percentage prevalence | | | Absolute prevalence | | |
|---|---|---|---|---|---|---|
| | Spearman rho (*vs.* call rate) | | Fisher combined p-value* | Spearman rho (*vs.* call rate) | | Fisher combined p-value* |
| | *Phase 1* | *Phase 2* | *Phases 1 and 2* | *Phase 1* | *Phase 2* | *Phases 1 and 2* |
| Complex | 0.07 | 0.27 | 0.4199 | 0.89 | 0.92 | **<0.0001** |
| Upward ramp | -0.14 | 0.23 | 0.4268 | 0.72 | 0.82 | **<0.0001** |
| Downward ramp | -0.13 | -0.26 | 0.3690 | 0.67 | 0.34 | **0.0004** |
| Flat | -0.44 | -0.11 | 0.9046 | 0.75 | 0.83 | **<0.0001** |
| Short | -0.50 | -0.37 | **0.0076** | 0.90 | 0.83 | **<0.0001** |
| Split | 0.06 | -0.01 | 0.9659 | 0.41 | 0.10 | 0.1420 |
| Step up | -0.09 | -0.18 | 0.6175 | 0.81 | 0.90 | **<0.0001** |
| Step down | -0.01 | -0.01 | 0.9977 | 0.88 | 0.91 | **<0.0001** |
| Multi-step | -0.44 | 0.15 | 0.0773 | 0.83 | 0.89 | **<0.0001** |
| Trill | 0.47 | 0.38 | 0.0102 | 0.87 | 0.81 | **<0.0001** |
| Flat-trill | 0.04 | -0.05 | 0.9535 | 0.81 | 0.70 | **<0.0001** |
| Trill with jumps | 0.46 | 0.24 | 0.0391 | 0.73 | 0.68 | **<0.0001** |
| Inverted-U | 0.08 | -0.24 | 0.4947 | 0.86 | 0.86 | **<0.0001** |
| Composite | 0.28 | 0.09 | 0.3908 | 0.69 | 0.25 | **0.0005** |

*Fisher combined p-values that reached significance (p < 0.01) are shown in bold.

subtypes. The first analysis investigated *relative* (i.e. percent) call prevalences. This analysis revealed no clear relationships between the 50-kHz call rate and the relative prevalence of individual call subtypes (Table 2, Fig 10), with the exception of short calls ($r_s$ = -0.50 and -0.37 for Phases 1 and 2, respectively, Fisher combined p = 0.0076).

In the second analysis, we determined correlations between the 50-kHz call rate and the *absolute* prevalence of each call subtype, with the latter now expressed as calls per minute; to avoid spurious positive correlations, the 50-kHz call rate was calculated separately for each subtype, after omitting the subtype under comparison. When prevalence was expressed in absolute terms, all subtypes were positively correlated with 50-kHz call rate (Table 2, S8 Fig), with Spearman rho values mostly lying between 0.70 and 0.90; all correlations were statistically significant, except for split calls (S8F Fig).

## Discussion

The present study, to our knowledge, provides the first description of 50-kHz call emission in a large group of adult rats over an extended period of repeated testing. The main novel findings are as follows. First, 50-kHz calling was maintained despite repeated testing in the same environment. Second, rats showed highly persistent, stable individual differences on our four main measures, i.e., percent sucrose preference, 50-kHz call rate, and percentage of flat and trill calls. Third, almost all rats preferentially emitted either flat or trill calls. Fourth, while 50-kHz call rate predicted the *absolute* emission rates of almost all call subtypes, it did not significantly predict the *relative* (i.e., percent) prevalence of any call subtype except for short calls. Notably, the 50-kHz call rate did not significantly predict the relative (i.e., percent) prevalences of flat and trill calls. Fifth, several correlations between specific call subtypes were revealed, including a reciprocal relationship between the percentages of flat and trill calls. Sixth, despite this reciprocal relationship, only flat calls were significantly correlated with a third call category made up of pooled non-trill/non-flat calls. Lastly, all USV measures examined were uncorrelated with SP.

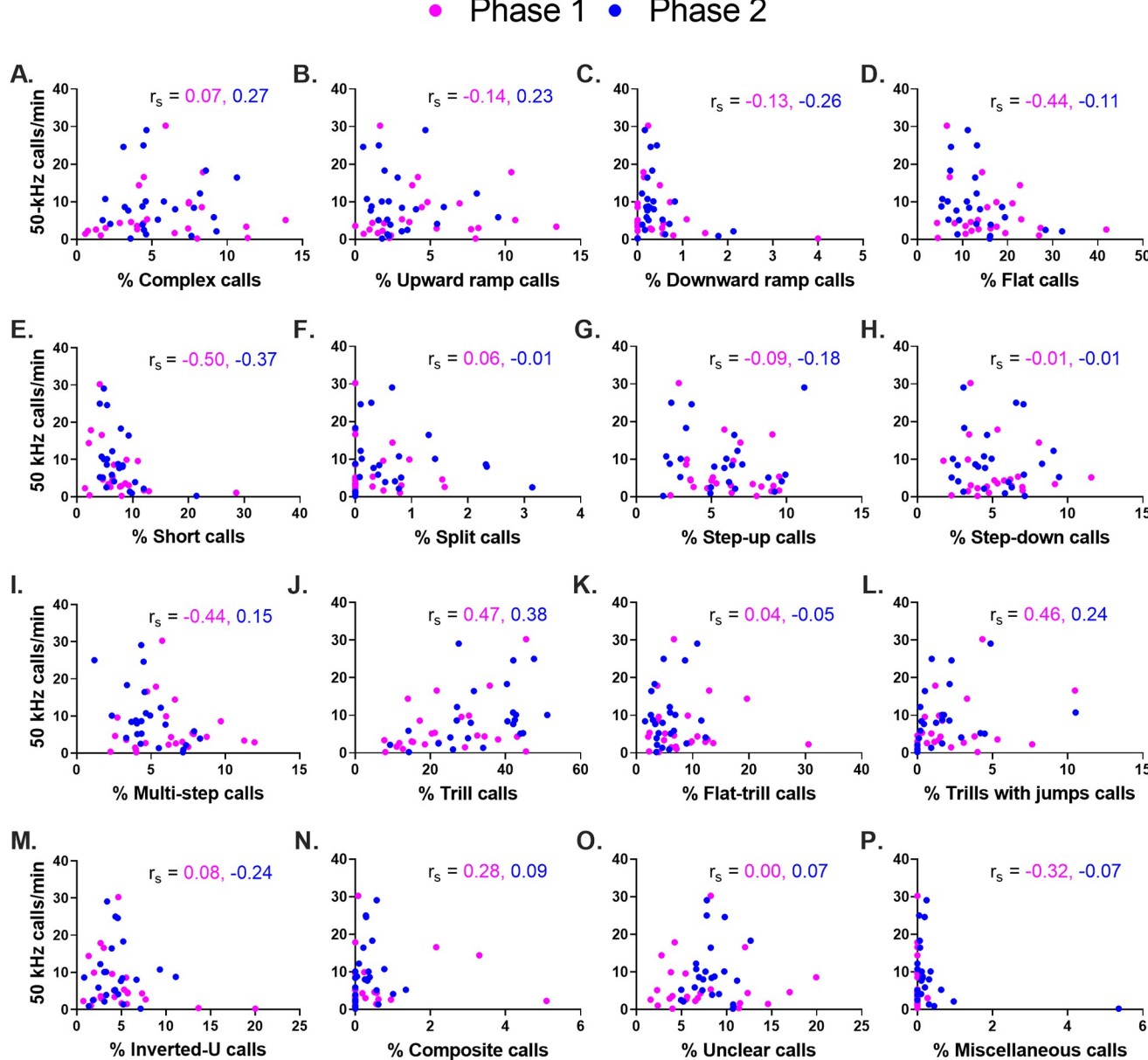

**Fig 10. The relationship between 50-kHz call rate and the relative prevalence of 50-kHz call subtypes.** (A-N) 50-kHz call rate *vs.* the percentage prevalence of each call subtype. Each symbol represents a single rat (n = 24 rats). Spearman correlation coefficients are shown within panels ($r_s$).

## Methodological considerations

The present USV testing conditions were relatively neutral, and differed substantially from those in previous studies where drugs or other (e.g. depressogenic) manipulations have been used. Since we needed to record individual animals, they were tested in an open field rather than the home cage; testing in the home cage would have meant temporarily removing the cage mate, or else subjecting all animals to long-term social isolation in individual cages. On USV test days, some disruption inevitably resulted from transport of rats to the test room, followed by their subsequent removal from the home cage and by their social separation from cage mates. This disruption was mitigated in two ways. First, subjects received two habituation

sessions prior to the first recording session, and second, rats were left to settle for at least 20 min upon arrival in the test room. Bedding is another potentially important variable that can affect USV emission [43, 44]. By replacing half the bedding in the test cages each day, we aimed to produce a familiar and reasonably stable olfactory setting from day-to-day. We consider half-replacement to be a better option than presenting the rats with 100% fresh bedding for each session, since transferring rats to cages with fresh bedding has been reported to trigger autonomic and behavioural arousal [45].

Our simulation-based correlational analyses offered several insights. Most importantly, when the percentage prevalences of different call categories were compared, the observed Spearman rho ($r_s$) values offered little guide to statistical significance. Evidently, in the extreme case of only two call categories, the $r_s$ value would always be exactly -1. With three call categories, our initial simulations still yielded strongly negative mean $r_s$ values based purely on random sampling. For example, even when simulating three call categories of equal prevalence (i.e. mean ± SD percentage prevalences: 33.3 ± 7 for each category), inter-category correlations averaged around -0.46 (n = 24 subjects, 1000 iterations sampled from a Gaussian distribution). With three call categories of *unequal* prevalence, a unique $r_s$ value was obtained for each inter-category comparison, and if two call categories predominated over the third category, then the associated $r_s$ value could be much closer to -1 (approaching the two-category case). This explains why in our study, the very strong negative correlations observed between trill and non-trill/non-flat categories failed to reach statistical significance (Table 1).

When the number of call categories was increased from 3 to 16, the simulated $r_s$ values now tended to distribute around zero. The same tendency was also seen in our observed data (mean ± SD of all Spearman rho coefficients for Phases 1 and 2, respectively: -0.04 ± 0.36 and -0.01 ± 0.31). Based on these findings, we initially intended to apply this simulation approach to the full range of call subtypes, in order to attribute a p-value to each of our observed intersubtype correlations. However, this approach was ultimately abandoned, since the lowest-prevalence call subtypes tended not to be normally distributed.

The present study employed a sucrose concentration (0.3%) that is lower than the 1–2% concentration range commonly used in animal models of depression [46, 47]. This 0.3% concentration was chosen because in pilot experiments, higher concentrations frequently resulted in SP scores close to 100%. Individual SP values covered a moderately wide range (i.e. 61%–86%), with a coefficient of variation (CV) of 0.10. In order to compare the latter with published CV values, we sampled around 20 SP studies that had used both a control group and a "depressogenic" procedure (S3 Table). Generally, calculated literature values tended to be higher (mean CV for control and "depressogenic" groups = 0.15 and 0.26, respectively). However, published studies typically featured a very limited number of SP test sessions, so it is unclear to what extent these higher CV values represented consistent inter-rat differences as opposed to random noise.

## Long-term individual differences in 50-kHz ultrasonic vocalizations

Individual differences in 50-kHz call rate have been widely reported, but with most reports concerned with stimulus-evoked, rather than non-evoked, conditions. In such studies, 50-kHz vocalizations have been induced under various conditions, e.g., tickling [19, 48], amphetamine administration [5, 16, 17, 31, 49, 50] and rough and tumble play [20, 51]. To our knowledge, only one study has demonstrated individual differences in 50-kHz call emission in the absence of an eliciting stimulus [14]. In this previous study, rats were tested in the home-cage environment for 4 days. Our study, which also used non-evoked conditions, demonstrated much longer-term individual differences in 50-kHz call rate, in a large group of animals.

The literature on individual differences in 50-kHz *call profiles* appears sparse. While temporal stability has been demonstrated in terms of group mean values [11, 20, 21, 30, 49], only Wright et al. (2010) have reported day-to-day variability of 50-kHz call profile in *individual rats*. However, the latter study was conducted under amphetamine administration, i.e., stimulus-evoked conditions, and for a short duration (6 test days).

Most of the 24 rats in the present study preferentially emitted trill calls, while almost all other rats preferred to emit flat calls. Subtype predominance does not appear to have been reported previously under non-evoked conditions, except in a preliminary (4-rat) study in which flat and short calls were prevalent [52]. Otherwise, the closest comparison from the literature would be saline challenge sessions, analyzed using the 14-subtype classification. Here, results vary between studies. In our own published work, we have noted trill calls [13, 53, 54] or flat calls [5, 9, 12] ranking as the predominant subtype, with one study reporting comparably high emission [55]. In contrast, Simola and coworkers have reported a predominance of flat and complex calls, with few trill calls emitted [50, 56]. This within- and between-laboratory variability may reflect differences in drug history or inter-batch differences, or other unidentified factors. In the wider literature examining stimulus-elicited USV emission, trill or flat calls have consistently appeared among the three most prevalent subtypes [20, 49, 57–61]. However, most of the latter studies used fewer 50-kHz call categories, and all of them used a calculation method giving more influence to higher-calling rats.

## 50-kHz call rate and prevalence of flat and trill calls

We next explored the relationship between 50-kHz call rate and call subtype, expressed both in absolute and relative terms. As expected, call rate was significantly and positively correlated with *absolute* prevalence for almost all call subtypes, i.e., all except split calls (mean ± SD of Spearman rho values: +0.77 ± 0.27 for Phase 1 and +0.70 ± 0.26 for Phase 2; S8 Fig). Importantly, each of these correlations was based on a 50-kHz call rate calculation that specifically excluded the call subtype in question; this method of calculation was used in order to prevent spurious positive correlations that might otherwise have arisen if the call rate from a given call subtype contributed to the overall call rate. These findings extend, and largely confirm, a previously-reported comparison of call rate and absolute call subtype prevalence [50]. However, our observed rho values were generally so high that the absolute prevalences provided little information beyond the 50-kHz call rate.

To our knowledge, the present study is the first to investigate a relationship between 50-kHz call rate and the *relative* (i.e., percentage) prevalence of call subtypes. Calculating prevalence in percentage terms offers two advantages: first, it is not influenced by manipulations that merely alter the call rate, and second, it is not disproportionately affected by high-calling subjects. In terms of relative prevalence, only one call subtype (short calls) was significantly correlated with the 50-kHz call rate. Of note, no relationship was detected between call rate and the percentage prevalence of trill or flat calls; hence, high-calling subjects were not necessarily trill-preferring or flat-preferring.

## Inter-call subtype correlations

Among 50-kHz call subtype relationships, we noted a striking and persistent inverse correlation between flat and trill calls; this relationship was clearly more marked than that expected through chance alone. A negative relationship was previously reported in a data set obtained from pooled amphetamine and vehicle test sessions (5), and is now reported under relatively neutral conditions as well as across several months. The present study also identified several inter-subtype relationships that to our knowledge have not been previously reported.

Since the relative prevalences of all 50-kHz call subtypes sums to 100%, it might be expected that most call subtypes would be negatively correlated. This tendency was strong in the case of only three call categories, but weak when the full range of sixteen 50-kHz call categories was considered (see Methodological considerations, above). Even controlling for this tendency, flat and trill calls still showed a clear reciprocal relationship. Our three-category analysis also showed that rats emitted trill calls only at the expense of flat calls, whereas they emitted flat calls at the expense of both trill calls and other call subtypes (the behavioural significance of this asymmetry is currently uncertain).

When the full range of 50-kHz call categories was included in the analysis, flat call prevalence showed statistically significant negative relationships not only with calls in the "unclear" category, but also with inverted-U and trill with jumps subtypes. Flat calls were *positively* associated with upward ramp and split calls, and while this finding might suggest that flat calls were most closely associated with other call subtypes possessing little frequency modulation, flat calls were not clearly correlated with either downward ramp or short calls. Trill call prevalence, in contrast, was positively correlated with the prevalence of inverted-U and unclear categories, with a similar trend for trill with jumps calls. Hence, it is tempting to speculate that inverted-U and trill with jumps call subtypes, which both have trill-like elements, may be functionally related to trill calls. Lastly, flat-trill combination calls were uncorrelated with either flat or trill calls.

## Stable individual differences in sucrose preference uncorrelated with USV measures

Although SP is a widely used measure, evidence for individual differences is largely indirect. Some relevant studies have associated a median split of SP with other behaviours [36, 62]. Three additional studies have shown consistent individual variability in SP, but only across 3–4 consecutive days [34, 39, 40]. We have now extended these findings using a larger group of rats, studied over a longer time frame (12 tests over 24 days). Individual pairs of tests tended to be only moderately predictive of SP values averaged across the remaining 5 day pairs (mean $r_s$ value ~ 0.4), with the first pair of tests being the least predictive. Thus, multiple daily test sessions would be highly desirable in order to generate reliable SP values for correlation with other behavioural measures.

In the present study, individual differences in 50-kHz *call rate* and SP were found to be unrelated under our relatively neutral conditions. To our knowledge, previously reported SP *vs.* call rate comparisons have concerned USV emission that was evoked—by social contact [63], tickling [19, 24, 31, 33], sucrose availability [32] or a chronic stress procedure [19, 31, 32]. Despite extensive testing, we were also unable to detect any relationship between emission of individual 50-kHz *call subtypes* and sucrose preference. The absence of detectable correlations between SP and USV-related measures is likely not due to low statistical power, for several reasons. First, Spearman rho values were generally low. Second, although our CV value was lower than in the average of a sample of published SP studies (see above, also S3 Table), our values would likely be less reflective of random noise as they are derived from a large number of test sessions per subject. Third, the correlation between SP and sucrose intake, in contrast, was close to statistical significance (p = 0.0118).

To our knowledge, only two studies have investigated a possible relationship between call subtype and SP [31, 63]. The first of these explored this relationship under depression-like conditions, reporting that both SP and emission of certain 50-kHz subtypes (trill, flat and short calls) were decreased by chronic variable stress. The second study reported greater stimulus-evoked emission of trill calls when preference was tested between lower *vs.* higher sucrose

concentrations [63]. Although the latter investigation was conducted under non-depressive like conditions, several procedural differences hinder comparison with the present study; notably, stimulus-evoked USVs were recorded in the presence of sucrose and a social partner.

The SP test is intended to capture anhedonia in rodents. However, depressed patients do not consistently show an analogous deficit [26–29]. Therefore, alternative affect-related measures are needed in animal models of depression. Accordingly, USV-related measures are starting to be used [19, 20, 31, 33]. In the present study, both SP and USV emission showed trait-like characteristics, but their lack of relationship suggests that USV recordings may capture a distinct marker of affect. It is as yet unclear, however, which particular USV-related measure (e.g. relative prevalence of flat or trill calls) may best track negative affect in these animal models.

## Strengths and limitations

The present study was unusually extensive in several aspects. First, we conducted long-term repeated testing of SP and USV emission in a large cohort comprising 24 rats. Second, a detailed analysis of 50-kHz calls was performed using the 14-call subtype scheme, thereby generating call profiles on many occasions for each individual subject. This study generated a large database of 50-kHz calls (~39,000 calls) which may prove useful for further data mining. Limitations include the following. A few rats showed consistent side preferences in the SP test, and this could have affected their SP scores even though water and sucrose bottle sides were counterbalanced within each day pair. Although basal USV emission would normally have been recorded in the home cage, we used a separate USV testing environment; this was to avoid the possible influence of sucrose-conditioned vocalizations. The present study used male adult Long-Evans rats, which were tested only under non-evoked conditions. Thus, it is as yet unclear whether our findings will be generalizable to other experimental contexts, e.g., different rat stocks, female rats, and other test conditions (e.g., different sucrose concentrations, stimulus-evoked USV emission).

## Conclusions

Long-term individual differences in USV emission appear unrelated to sucrose preference, strengthening the case for investigating USV-related measures in animal models of depression. More generally, if USV measures were to be applied to the routine surveillance of animal welfare, the existence of marked and temporally-stable individual differences would need to be taken into account.

## Supporting information

**S1 Table. Spearman rho ($r_s$) correlation coefficients relating sucrose preference to the relative prevalence of 50-kHz call subtypes in Phase 1 (no significant correlations).** SP and USVs were *both* measured in Phase 1 only. The relative (i.e., percentage, not absolute) prevalence of the 50-kHz call subtypes were used in the analysis. Spearman rho critical values for n = 24 are: 0.409 for p<0.05, and 0.537 for p<0.01 (2-tailed). The alpha level was set at p < 0.01 (2-tailed).
(DOCX)

**S2 Table. Spearman rho ($r_s$) correlation coefficients and Fisher combined p-values for inter-call subtype relationships.** The relative (i.e., percentage, not absolute) prevalence of the 50-kHz call subtypes were used in the analysis. The alpha level was set at p < 0.01 (2-tailed).
(DOCX)

**S3 Table. Coefficient of variation values for sucrose preference tests.** The coefficient of variation was calculated for 23 studies in the literature, for both the "stressed" and control groups. The mean, SD and N were stated in the text of the study, or could be estimated from a figure. (DOCX)

**S1 Fig. Individual differences in 50-kHz USV measures in Phase 1.** The 24 rats were median-split into low and high subgroups (n = 12), based on the following measures: (A and B) 50-kHz call rate, (C and D) percentage of flat calls, and (E and F) percentage of trill calls. Each line represents an individual rat and each symbol represents a day pair (6 day pairs). (TIF)

**S2 Fig. Ranked individual differences in 50-kHz USV measures in Phase 1.** Rats were split into high and low performers based on a median split of each of the three main USV measures. Rats were then ranked on their performance on each measure, for each day pair (1 = lowest performer, 24 = highest performer). (A and B) The ranked 50-kHz calling rate is shown for each rat across the 6 day pairs. Ranked percentage prevalence of (C and D) flat calls and (E and F) trill calls were very stable across the 6 day pairs. Each symbol represents a day pair and each line represents a rat. (TIF)

**S3 Fig. Individual differences in 50-kHz USV measures in Phase 2.** A median split of each USV measure was performed to separate rats into high and low performers. (A and B) The rate of 50-kHz calling is shown for each rat across the 3 day pairs. Percentage of (C and D) flat calls, (E and F) and trill calls (E and F) across the 3 day pairs. Each symbol represents a day pair and each line represents a rat. (TIF)

**S4 Fig. Ranked individual differences in 50-kHz USV measures in Phase 2.** A median split of each USV measure was performed to separate rats into high and low performers. Rats were then ranked on their performance on each measure, for each day pair (1 = lowest performer, 24 = highest performer). (A and B) The ranked 50-kHz calling rate is shown for each rat across the 3 day pairs. Ranked percentage prevalence of (C and D) flat calls, and (E and F) trill calls were very stable across the 3 day pairs. Each symbol represents a day pair and each line represents a rat. (TIF)

**S5 Fig. 50-kHz call subtype profiles for each rat in Phase 2.** The mean + SEM percentage prevalence of each call subtype is shown (n = 3 day pairs). Most rats show a preference for trill calls over flat calls. Red arrows point to the prevalence of flat and trill calls. Call subtype definitions are as follows; CX: complex, UR: upward ramp, DR: downward ramp, FL: flat, SH: short, SP: split, SU: step-up, SD: step-down, MS: multi-step, TR: trill, FT: flat-trill, TJ: trill with jumps, IU: inverted-U, CS: composite, UC: unclear, MI: miscellaneous. (TIF)

**S6 Fig. Individual differences in sucrose preference and intake in Phase 1.** Each rat was tested in 12 test sessions and data were averaged to form 6 day pairs (each symbol represents a day pair and each line represents a rat). The 24 rats were median-split into low and high subgroups (n = 12), based on their percent sucrose preference. Panels A and B show sucrose preference, whereas panels C and D show sucrose intake. Panels E-H show the same data as in panels A-D, except that y-axis values show the *ranked* score for each rat on a given day pair, for sucrose preference (panels E and F) or sucrose intake (panels G and H). For each day pair, subjects were assigned a rank based on their sucrose preference and sucrose intake (1 = lowest

performer, 24 = highest performer).
(TIF)

**S7 Fig. Individual differences in sucrose intake versus main variables of interest.** (A) Non-significant trend relating sucrose preference to sucrose intake. (B-D) the lack of relationship between sucrose intake and the three main USV measures, i.e., (B) call rate, percentage of (C) flat calls, (D) percentage of trill calls. Each symbol represents a single rat (n = 24 rats). Spearman correlation coefficients are shown within panels ($r_s$).
(TIF)

**S8 Fig. The relationship between 50-kHz call rate and the absolute prevalence of 50-kHz call subtypes (i.e., calls/min).** (S8A-P) 50-kHz call rate was calculated exclusively for each subtype by excluding that subtype from the analysis (i.e., using the remaining subtypes). A positive correlation can be observed between 50-kHz call rate and almost all 50-kHz call subtypes. Each symbol represents a single rat (n = 24 rats). Spearman correlation coefficients are shown within panels ($r_s$).
(TIF)

## Acknowledgments

We wish to thank lab members Emma Paulus, Hanan Mohammad, and Lucas Marques for providing comments on the manuscript. P. Clarke is also a member of the Center for Studies in Behavioral Neurobiology at Concordia University, Montreal.

## Author Contributions

**Conceptualization:** Adithi Sundarakrishnan, Paul B. S. Clarke.

**Data curation:** Paul B. S. Clarke.

**Formal analysis:** Adithi Sundarakrishnan, Paul B. S. Clarke.

**Methodology:** Adithi Sundarakrishnan.

**Project administration:** Paul B. S. Clarke.

**Supervision:** Paul B. S. Clarke.

**Writing – original draft:** Adithi Sundarakrishnan.

**Writing – review & editing:** Adithi Sundarakrishnan, Paul B. S. Clarke.

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
