## [Decision Letter · Decision Letter 0]

13 Jun 2022

PONE-D-22-12721Stable long-term individual differences in basal 50-kHz vocalization rate and call subtype prevalence in adult male rats: comparisons with sucrose preferencePLOS ONE

Dear Dr. Clarke,

Thank you for submitting your manuscript to PLOS ONE. After careful consideration, we feel that it has merit but does not fully meet PLOS ONE’s publication criteria as it currently stands. Therefore, we invite you to submit a revised version of the manuscript that addresses the points raised during the review process.

We look forward to receiving your revised manuscript.

Kind regards,

Brenton G. Cooper, Ph.D.

Academic Editor

PLOS ONE

Journal Requirements:

2. Please provide additional details regarding animal euthanasia in the body of your manuscript.

Reviewers' comments:

Reviewer's Responses to Questions

**Comments to the Author**

1. Is the manuscript technically sound, and do the data support the conclusions?

Reviewer #1: Yes

Reviewer #2: Yes

2. Has the statistical analysis been performed appropriately and rigorously? 

Reviewer #1: Yes

Reviewer #2: No

3. Have the authors made all data underlying the findings in their manuscript fully available?

Reviewer #1: No

Reviewer #2: Yes

4. Is the manuscript presented in an intelligible fashion and written in standard English?

Reviewer #1: Yes

Reviewer #2: Yes

5. Review Comments to the Author

Reviewer #1: As the manuscript stands, there is no linked access to the data or statement indicating where / how to access the data.

The objectives of the indicated study are to observe, in rats, the relationship between Ultrasonic Vocalization (USV) content and sucrose preference (SP) over time under basal conditions (no intervention). Both USV and SP are evidenced to provide information relevant to affect; however, it is uncertain if they are of unique affective dimensions. Further consideration is given to the individual stability of USV and sucrose preference, with further investigation into USV sub-type components. The paper is well-researched and flows well. The statistical analysis is well described, however, there are a couple elements that require further clarification.

Reviewer #2: This study sought to test whether vocalizations under relatively neutral conditions are stable across time and, more specifically, whether the relative prevalence of each of many different types of calls is characteristic of an individual rat across a time frame of weeks to months. It also sought to test whether sucrose preference, a common measure of anhedonia in rodents, was stable across time and whether this measure was related to individual differences in call profiles, by which I mean the relative prevalence of each call subtype. The findings support the idea that the call profile while in an empty box is different for each animal and relatively stable across time. Sucrose preference was also found to exhibit individual differences which were stable across time but there was no correlation between sucrose preference and 50 kHz vocalization rate nor any of the sub-types of vocalizations. The paper also offers a rich array of exploratory analyses of the relationships between overall call rates and call profiles as well as relationships between the prevalence of individual calls. While these latter analyses offer some potentially interesting findings, I have serious concerns about the analysis methods, so it is impossible to assess the importance of these findings at this point.

This paper will be of interest to those in the field of affect/vocalization studies in rats and is likely to be cited to support the utility of vocalization profiles in studies of individual differences. In my estimation, this paper’s most significant findings are: 1) The relative stability of call subtype prevalence across days and even weeks. 2) Lack of correlation of sucrose preference, a standard measure of anhedonia, with calls, which many in the field have assumed are indicators of affective state in rats. This latter finding is particularly striking and definitely worth reporting. The study is methodologically sound and is admirable for the particularly large sample size. On the other hand, I had significant concerns about several of the analyses, as detailed below. Further, I disagree with the authors characterization of their testing state as basal (also discussed further below). I’ve listed other concerns, both major and minor, below.

Line 207: Excluding values greater than 10X the mean is not a method I’ve seen before to define outliers. If there is precedence, please cite a reference. Otherwise, it would be preferable to use a standard method to avoid the appearance of experimenter subjective bias (why choose 10X? Why not 5X?). A more standard method is to use the median absolute deviation (MAD) (https://doi.org/10.1016/j.jesp.2013.03.013).

Line 208: It would be preferable to compute your statistics using missing values rather than replacing the missing values with the mean from all other sessions. I’m sure the effect is slight in this case, but technically, this does artificially increase your power and hence increase the chance of a false positive test result.

Lines 214-216: A bit more specificity, either in the methods or results, would be helpful as to how the Kruskall-Wallis test and Cronbach’s alpha values were computed. For example, it took me a while to figure out that Cronbach’s alpha was used to determine similarity of the 6 day-pair vocalization rates to each other.

Line 261: When correlating each day-pair against the mean, did you exclude that day-pair from the currently calculated mean? Failure to do so will introduce bias into your analysis as you are essentially correlating a measure with another measure of which is a significant part. I ran a quick numerical simulation of this scenario with random data and got greater than 25% false positives. Same issue applies to SP data (line 337).

Fig. 5: The negative relationship between trill and flat calls in 5E-F may be misleading because they are expressed as proportions. To illustrate, consider the case where you only had trill and flat calls. If 60% of your calls are trill, 40% would have to be flat. With 90% trills, you’d have 10% flat and so on. There are other calls, so the situation is not quite as dire as illustrated, but the problem is still significant. To avoid this, look for a relationship between absolute trill and flat call rates. Also, these two panels (E-F) should be mentioned in the results section.

Line 292 & Fig 4A: Are phase 1 and phase 2 call rates statistically different?

Fig. 6: I don’t see the need for panels 6A, 6C, and 6E. The information plotted is the same. Panels A, C and E emphasize group differences between Phase 1 and 2 but panels B, D, and F emphasize the similarity in measures, which, I believe, is the point you are trying to make.

Line 367-393: Correlations based on proportions will be biased toward negative correlations. Use raw call rates instead. See my comments on Fig. 5.

Fig. 10 Panel O,P: See my comments about correlating proportions on Fig. 5.

Line 404-401 & Table 1 & Fig. S8: The number of a specific call contributes to the overall number of calls. Hence, these two numbers will tend to be correlated, even in random data. Its like showing that A is correlated with (A+B+C+D). For this analysis of the relationship between call subtype prevalence and call rate, best to stick with the proportion.

Line 444: “rats were tested in the home-cage environment for 4 days”. I did not see reference to home-cage USV data in methods or results.

Discussion (Line 439-446 and elsewhere): The discussion and intro stress that this is the first report of “basal” or “spontaneous” vocalizations and contrast these with other experiments which have studied vocalizations elicited by specific experiences, such as social contact, trickling, or sucrose. To me, basal recordings would be conducted in the home cage, when nothing is changed about the rat’s experience. In the present study, rats are 1) moved to a novel non-homecage environment 2) exposed to scents from unfamiliar rats (previous occupants of shared testing chamber) and 3) separated from their constant companion. Habituation might arguably reduce the impact of the novelty but the longing for a companion is likely to maintain across the experiment and, based on the work of Markus Wohr and others, is likely to influence vocalizations. (Wohr has shown that separating a rat from others causes the emission of 50 kHz short calls: Wohr, M. et al. 2008. Physiol Behav 93, 766-776). I would describe your testing conditions are “relatively neutral” but whatever you choose to call it, it is worth some discussion of some of the factors, especially social factors, which may influence calling under these conditions. This issue is mentioned briefly on line 547, but deserves more consideration.

Line 504-535: One possible explanation for your lack of effect in the vocalization vs SP correlation may be a lack of range in the SP measurements. Lacking a treatment, all rats have SP in the “normal” range. Studies, such as Ref. 33, which have a manipulation which increases or decreases depressive-like behavior might have found relationships between SP and vocalizations because the manipulations creates two groups of animals, some with low SP and other with high SP. This separation in the data adds power to their analysis. Looking at some of the studies, it appears to me that the spread in individual differences in SP in the present study is comparable to that induced by some of these manipulations, but it is worth considering this possibility explicitly in the discussion.

Line 515: Ref 33 actually used tickling, not social play to elicit vocalizations.

6. PLOS authors have the option to publish the peer review history of their article (what does this mean?). If published, this will include your full peer review and any attached files.

Reviewer #1: **Yes: **Logan J. Bigelow

Reviewer #2: **Yes: **David R. Euston

---

## [Author Response · Author response to Decision Letter 0]

30 Jul 2022

We warmly thank the two reviewers for their careful reading of the manuscript and for their insightful and detailed critiques. We have responded to every point, as indicated by "AUTH" in the text below (line numbers refer to the revised manuscript). As indicated below, we are in agreement with all the substantive comments. 

During MS revision, we noticed some errors in the data analysis, which have now been corrected, as follows.

First, some of the calls for rat 19 on day 2 were inadvertently duplicated in our datasheet (i.e. ~180 calls). These extraneous calls made up only a small portion of our dataset (180/24,456 calls in Phase 1). After deleting these duplicated calls, we re-ran all the analyses and updated the figures. This reanalysis did not change our results significantly in any respect (if you wish to check, please consult the document that has the track changes). 

Second, we noticed a calculation error in the data presented in Fig 3D. We have now corrected this error, both in panel 3D and also in the Results (lines 322-325). This means that in Phase 2, we now report a statistically significant inter-individual difference for the third main USV-related measure (percent flat calls). This slightly strengthens the overall conclusions of the study.

Finally, we have made quite a number of minor edits, mainly to improve clarity or flow.

REVIEWER #1

As the manuscript stands, there is no linked access to the data or statement indicating where / how to access the data.

AUTH: Our Data Availability statement now reads: "Our dataset of acoustic (WAV) files is very large (800+ files, totalling approx. 400 GB). We will share these WAV files upon request. The numerical data derived from these WAV files, together with the raw data from SP tests, can be found at 10.6084/m9.figshare.20341206."

The objectives of the indicated study are to observe, in rats, the relationship between Ultrasonic Vocalization (USV) content and sucrose preference (SP) over time under basal conditions (no intervention). Both USV and SP are evidenced to provide information relevant to affect; however, it is uncertain if they are of unique affective dimensions. Further consideration is given to the individual stability of USV and sucrose preference, with further investigation into USV sub-type components. The paper is well-researched and flows well. The statistical analysis is well described, however, there are a couple elements that require further clarification. 

Minor Comments 

Line 17: Is it sucrose preference or sucrose preference test, be consistent 

AUTH: Throughout the manuscript, "SP" refers to sucrose preference. When the SP test is meant, we have tried to ensure that the word "test" is included.

Line 31-33: A greater distinction needs to be made here. The initial statement regarding measures of interest include rate of 50 kHz calls, and relative prevalence of trill and flat call subtypes. 

AUTH: We are uncertain what this comment means. Please clarify if it is considered important.

Line 43: Reflected.

AUTH: We feel that "reflect" is the correct tense in this case ("It was initially proposed that 22-kHz calls reflect negative affect").

Line 46: You need to provide a bit of background on the classification schematic.

AUTH: Unfortunately, we do not know what this reviewer is asking for. The 14-call subtype classification schematic is fairly well known, with at least a dozen labs around the world using it, either in whole or in modified form. 

Line 47-48: Respectively?

AUTH: Fixed (line 44).

Line 81-82: Reword: “has been little investigated”

AUTH: Fixed (line 77-78).

Line 106: Were the animals handled prior to experimentation? If so, how much and how long? Were the animals allowed acclimation to the facility prior to the beginning of the experiment?

AUTH: Thank you for pointing out these missing details. Animals were indeed handled and allowed to acclimate, as now described (lines 104-108). 

Line 108-109: How old? Days? This is important for studies concerning USV emission and replicability.

AUTH: These missing details have been added (lines 107-108).

Line 124: Either take out 30 sessions or state 30 sessions: 12 SP, 18 USV.

AUTH: Fixed (line 123).

Line 138: Was there any reason you decided on 13 weeks?

AUTH: It was simply a conveniently long period.

Line 145-146: Any bedding in these cages?

AUTH: Yes. This detail has now been added (line 109).

Line 178: Did you assess whether USV in one open field could be detected in another? 

AUTH: Yes, we did. We used a custom-made ultrasound emitter that generated trains of 50-kHz pulses of variable intensity. This allowed us to verify that 50-kHz calls would be detectable from all locations within a given test box, but not by microphones in neighbouring test boxes. This is now stated (lines 192-195).

Where did the testing occur? SP took place in the colony room, did USV as well?

AUTH: USV testing took place in a different room, as now stated (line 175).

Line 179: Why change half? Are you attempting to provoke USV by scented bedding?

AUTH: We realize that bedding is an important consideration in USV experiments (Brudzynski and Pniak 2002; Natusch and Schwarting 2010). Our aim was not to provoke USV emission. Rather, we aimed to provide a familiar and relatively stable olfactory environment from day to day; this is now explained on line 468-473. To have replaced soiled bedding with fresh bedding before every test session would also have been laborious, as we were testing 24 rats per day. Also, please note that the animals also received habituation sessions before Phase 1 and again before Phase 2.

Line 188: Please state the frequency cutoff limit for 50 vs 22 kHz calls. What was the minimum call intensity?

AUTH: Frequency cutoffs have been added (line 199-200). As in our previous publications, USVs were visually identified in spectrograms and no call intensity threshold was applied; we now state this explicitly (line 198-199). 

Line 191-192: Explicitly state the length of the trial somewhere above. How long were the habituation trials for USV?

AUTH: All habituation and test sessions were of 20 minutes duration, as now stated on line 176.

Line 212-213: For the SP, data was removed because it affected results, but it was not removed for the USV because it didn't affect the results. Decide whether outliers will be removed, and on what basis, and then run the analysis. The decision should be made and followed through with before any analysis is completed.

AUTH: Please see our response to Reviewer #2 on this same point.

Line 245: Add a comma or "as well as" 

AUTH: We have added the comma (line 268).

Line 302: Respectively?

AUTH: Fixed. We agree "respectively" is a better word (line 324).

Line 305: I would suggest taking out the subheadings of phase 1 and phase 2 as most of this is a comparison between the two phases. I think it would also help the flow of the paper to discuss/explain the figures in order highlighting the similarities and differences between the two phases.

AUTH: We have retained the separate subheadings for Phases 1 and 2, but we have clarified the text where Phase 1 and 2 are compared by adding a new subheading, i.e. "Phase 1 vs. 2."

AUTH: Indeed, while reading the Results text, the reader has to jump back several times in order to refer to an earlier multi-panel figure. Unfortunately, we have found no way to avoid this. A key theme of our study is the demonstration of both shorter-term vs. long-term individual differences, so we need to report results from the two phases separately before comparing them.

Line 356: Write A, B, C after what each describes.

AUTH: Fixed (line 381-383).

Line 405: Is the difference between absolute and relative that absolute is the number of calls and relative is the value as a percent of total calls? It would be helpful to define this in the methods and clarify.

AUTH: That is correct. We have clarified this in the Statistical analysis section, on lines 215-222.

Lines 475-477: An explanation of how relative prevalence was calculated would be helpful in understanding how it is not skewed by high calling rats (this would be beneficial in the methods). 

AUTH: We have clarified our methods of calculation, and we have also inserted the following sentence: "As a rate-free measure, relative (i.e. percent) prevalence is not skewed by high-calling rats." Please see lines 218-222.

REVIEWER #2 

Reviewer #2: This study sought to test whether vocalizations under relatively neutral conditions are stable across time and, more specifically, whether the relative prevalence of each of many different types of calls is characteristic of an individual rat across a time frame of weeks to months. It also sought to test whether sucrose preference, a common measure of anhedonia in rodents, was stable across time and whether this measure was related to individual differences in call profiles, by which I mean the relative prevalence of each call subtype. The findings support the idea that the call profile while in an empty box is different for each animal and relatively stable across time. Sucrose preference was also found to exhibit individual differences which were stable across time but there was no correlation between sucrose preference and 50 kHz vocalization rate nor any of the sub-types of vocalizations. The paper also offers a rich array of exploratory analyses of the relationships between overall call rates and call profiles as well as relationships between the prevalence of individual calls. While these latter analyses offer some potentially interesting findings, I have serious concerns about the analysis methods, so it is impossible to assess the importance of these findings at this point.

This paper will be of interest to those in the field of affect/vocalization studies in rats and is likely to be cited to support the utility of vocalization profiles in studies of individual differences. In my estimation, this paper’s most significant findings are: 1) The relative stability of call subtype prevalence across days and even weeks. 2) Lack of correlation of sucrose preference, a standard measure of anhedonia, with calls, which many in the field have assumed are indicators of affective state in rats. This latter finding is particularly striking and definitely worth reporting. The study is methodologically sound and is admirable for the particularly large sample size. On the other hand, I had significant concerns about several of the analyses, as detailed below. Further, I disagree with the authors characterization of their testing state as basal (also discussed further below). I’ve listed other concerns, both major and minor, below.

Line 207: Excluding values greater than 10X the mean is not a method I’ve seen before to define outliers. If there is precedence, please cite a reference. Otherwise, it would be preferable to use a standard method to avoid the appearance of experimenter subjective bias (why choose 10X? Why not 5X?). A more standard method is to use the median absolute deviation (MAD) (https://doi.org/10.1016/j.jesp.2013.03.013).

AUTH: We have rewritten this section (lines 223-228). The previous "at least 10-fold greater than the median value costs" description was chosen arbitrarily, and was merely intended to highlight how really extreme these sucrose or water intake values were, especially given that the rats were not water-deprived. Although not previously stated, our motivation for excluding these 3 extreme values is that they are almost certainly due to fluid leakage, and hence erroneous. To be 100% sure, we should have checked for damp bedding, but this was not done. These few very high values were not associated with rats that naturally drank the most, and by way of comparison, non-deprived male rats of comparable weight would typically be expected to consume 18-50 ml in a 24-h period, not in a 1-h test session (1, 2). 

Line 208: It would be preferable to compute your statistics using missing values rather than replacing the missing values with the mean from all other sessions. I’m sure the effect is slight in this case, but technically, this does artificially increase your power and hence increase the chance of a false positive test result.

AUTH: Thank you for suggesting this approach, which we have adopted, i.e., we now treat these extreme fluid intake values as missing. This has changed the results of three types of analysis involving SP (sucrose preference), and as expected in each case the change was slight: (1) The Cronbach's alpha value for SP increased from 0.59 to 0.61, (2) the Kruskal-Wallis result (Fig. 3G) showing individual differences has become more significant, i.e. p = 0.0037 became p = 0.0024, and (3) one of the Spearman correlation coefficients relating SP to the USV measures (specifically, the percentage of trill calls) has changed slightly i.e. rho -0.17 became -0.18. The other correlation coefficients did not change, reflecting the fact that Spearman correlation is based on ranked rather than absolute values.

Lines 214-216: A bit more specificity, either in the methods or results, would be helpful as to how the Kruskall-Wallis test and Cronbach’s alpha values were computed. For example, it took me a while to figure out that Cronbach’s alpha was used to determine similarity of the 6 day-pair vocalization rates to each other.

AUTH: We have added more details, please see lines 234-238.

Line 261: When correlating each day-pair against the mean, did you exclude that day-pair from the currently calculated mean? Failure to do so will introduce bias into your analysis as you are essentially correlating a measure with another measure of which is a significant part. I ran a quick numerical simulation of this scenario with random data and got greater than 25% false positives. Same issue applies to SP data (line 337).

AUTH: First, just as a reminder, as per the previous suggestion, we have treated the extreme fluid intake values as missing values, and we have re-performed this analysis.

AUTH: In our original submission, we wanted to show how well each individual day pair represented the mean of all six pairs of days. This is why the day pair itself was included in the calculation of the average. Thank you for pointing out that this approach would bias the results towards a positive correlation. Accordingly, we have now replaced our analyses of SP and USVs with the analysis that you recommended, i.e. correlating a given day pair with an averaged value that specifically excludes the day pair being compared. These new results are on lines 284-289 (for Phase 1 USV measures) and on lines 362-365 (for the SP measure). This has not changed the overall conclusion: the first day pair is still the least representative.

Fig. 5: The negative relationship between trill and flat calls in 5E-F may be misleading because they are expressed as proportions. To illustrate, consider the case where you only had trill and flat calls. If 60% of your calls are trill, 40% would have to be flat. With 90% trills, you’d have 10% flat and so on. There are other calls, so the situation is not quite as dire as illustrated, but the problem is still significant. To avoid this, look for a relationship between absolute trill and flat call rates. Also, these two panels (E-F) should be mentioned in the results section.

AUTH: Indeed, when call subtypes are expressed in percentage (rather than absolute) terms, then for mathematical reasons one might well expect a reciprocal relationship (negative correlation) between trill and flat calls, or perhaps more generally between other pairs of 50-kHz call subtypes. However, this negative relationship is surprisingly limited in extent. We have extended our analysis relating to this point, please see Discussion lines 532-544. We have also moved up the subheading "Inter-call subtype correlations" (new line number 524) so that it includes discussion of these points.

AUTH: As suggested, we have also analyzed the relationship between call subtypes in terms of absolute call rates. This analysis yielded an overwhelming preponderance of positive correlations, which is most likely driven by the large individual differences in 50-kHz call rates (thus, a rat that makes a high number of one call subtype will tend to make a high number of all other call subtypes). Please see Discussion lines 538-544.

Line 367-393: Correlations based on proportions will be biased toward negative correlations. Use raw call rates instead. See my comments on Fig. 5.

AUTH: Please see response immediately above.

Fig. 10 Panel O,P: See my comments about correlating proportions on Fig. 5.

Correlate absolute prevalence of call subtypes – with absolute call rate

AUTH: Please see response immediately above.

Line 404-401 & Table 1 & Fig. S8: The number of a specific call contributes to the overall number of calls. Hence, these two numbers will tend to be correlated, even in random data. Its like showing that A is correlated with (A+B+C+D). For this analysis of the relationship between call subtype prevalence and call rate, best to stick with the proportion.

AUTH: Please see response further above.

Line 292 & Fig 4A: Are phase 1 and phase 2 call rates statistically different?

AUTH: Thank you for pointing out this omission. Rats tended to emit fewer 50-kHz calls in Phase 2 than in Phase 1, but this did not reach our 1% significance threshold. We now include this and other comparisons between the two phases, i.e. not only for 50-kHz call rate but also for the percentage of flat and trill calls. Please see the new subsection called "Phase 1 vs. 2", lines 330-333. 

Fig. 6: I don’t see the need for panels 6A, 6C, and 6E. The information plotted is the same. Panels A, C and E emphasize group differences between Phase 1 and 2 but panels B, D, and F emphasize the similarity in measures, which, I believe, is the point you are trying to make.

AUTH: Agreed. We have removed these panels.

Line 444: “rats were tested in the home-cage environment for 4 days”. I did not see reference to home-cage USV data in methods or results.

AUTH: We feel that our statement is justified. Specifically, the Schwarting et al. 2007 paper cited in this context (https://doi.org/10.1016/j.bbr.2007.01.029) included a group of rats that were recorded in their home cages (albeit in a separate testing room) - please see their Methods section 2.1.2 and also their Figs. 2-4. 

Discussion (Line 439-446 and elsewhere): The discussion and intro stress that this is the first report of “basal” or “spontaneous” vocalizations and contrast these with other experiments which have studied vocalizations elicited by specific experiences, such as social contact, trickling, or sucrose. To me, basal recordings would be conducted in the home cage, when nothing is changed about the rat’s experience. In the present study, rats are 1) moved to a novel non-homecage environment 2) exposed to scents from unfamiliar rats (previous occupants of shared testing chamber) and 3) separated from their constant companion. Habituation might arguably reduce the impact of the novelty but the longing for a companion is likely to maintain across the experiment and, based on the work of Markus Wohr and others, is likely to influence vocalizations. (Wohr has shown that separating a rat from others causes the emission of 50 kHz short calls: Wohr, M. et al. 2008. Physiol Behav 93, 766-776). I would describe your testing conditions are “relatively neutral” but whatever you choose to call it, it is worth some discussion of some of the factors, especially social factors, which may influence calling under these conditions. This issue is mentioned briefly on line 547, but deserves more consideration.

AUTH: We wish to emphasize that our testing conditions are substantially different from previous studies which have used drugs or other (e.g. depressogenic) manipulations to alter USV emission. The challenge is, what term to use? We have now abandoned the term "basal" (which might prompt the question: as a baseline to what?), including from the title. We have also avoided the term "spontaneous" (which would imply the absence of any external stimulus). Thank you for suggesting the term "relatively neutral", which we have adopted in several places. We also occasionally refer to "non-evoked" conditions.

AUTH: This reviewer raises some important points that we agree deserved more discussion. To this end, we have now added a new section early in the Discussion called "Methodological considerations" (lines 460-478). We now point out that removing a test rat from its cage mate produces social isolation. However, we have not mentioned the Wöhr et al 2008 findings, as we consider that they do not convincingly show that separating a rat from its cage mate caused the emission of 50 kHz calls. To demonstrate this, their experimental designs would have needed a control group of subjects that were not socially separated (Experiment A, housing cage test, section 2.1.4; Experiment B section 3.1.2). By the way, Wöhr et al referred to increased emission of flat and frequency-modulated calls rather than short calls per se. 

Line 504-535: One possible explanation for your lack of effect in the vocalization vs SP correlation may be a lack of range in the SP measurements. Lacking a treatment, all rats have SP in the “normal” range. Studies, such as Ref. 33, which have a manipulation which increases or decreases depressive-like behavior might have found relationships between SP and vocalizations because the manipulations creates two groups of animals, some with low SP and other with high SP. This separation in the data adds power to their analysis. Looking at some of the studies, it appears to me that the spread in individual differences in SP in the present study is comparable to that induced by some of these manipulations, but it is worth considering this possibility explicitly in the discussion.

AUTH: Thank you for raising these points. Most are now covered either in our new Discussion section "Methodological considerations" (lines 475-478) or later in the Discussion (lines 573-579). We too consider that our SP values covered a fairly large range (i.e. 61% - 86% between rats), which is advantageous for correlational analysis. This range reflects our choice of a relatively low sucrose concentration (0.3%). Although higher concentrations of 1-2% are standard in animal models of depression (Strekalova et al 2022 Table 2, Scheggi et al 2018), pilot testing revealed that in our hands, concentrations above 0.3% frequently resulted in SP scores close to 100%.

AUTH: Although in the present study SP did not appear related to any USV-related measure, this is likely not due to low statistical power, for several reasons (now added on lines 570-576). First, Spearman rho values were generally low. Second, there was substantial inter-rat variability in both SP and USV measures. Third, the correlation between SP and sucrose intake, in contrast, was close to statistical significance (p = 0.0118). 

Line 515: Ref 33 actually used tickling, not social play to elicit vocalizations.

AUTH: Fixed (lines 75 and 570) – thanks for catching this.

---

## [Decision Letter · Decision Letter 1]

24 Aug 2022

PONE-D-22-12721R1Stable long-term individual differences in 50-kHz vocalization rate and call subtype prevalence in adult male rats: comparisons with sucrose preferencePLOS ONE

Dear Dr. Clarke,

Thank you for submitting your manuscript to PLOS ONE. After careful consideration, we feel that it has merit but does not fully meet PLOS ONE’s publication criteria as it currently stands. Therefore, we invite you to submit a revised version of the manuscript that addresses the points raised during the review process.  I have taken time to review the statistical issues raised by one of the reviewers and agree that this issue needs to be addressed prior to a decision being made regarding suitability for publication.  Please address these issues and we look forward to evaluating your response.

We look forward to receiving your revised manuscript.

Kind regards,

Brenton G. Cooper, Ph.D.

Academic Editor

PLOS ONE

Journal Requirements:

Reviewers' comments:

Reviewer's Responses to Questions

**Comments to the Author**

1. If the authors have adequately addressed your comments raised in a previous round of review and you feel that this manuscript is now acceptable for publication, you may indicate that here to bypass the “Comments to the Author” section, enter your conflict of interest statement in the “Confidential to Editor” section, and submit your "Accept" recommendation.

Reviewer #1: All comments have been addressed

Reviewer #2: (No Response)

2. Is the manuscript technically sound, and do the data support the conclusions?

Reviewer #1: Yes

Reviewer #2: Yes

3. Has the statistical analysis been performed appropriately and rigorously? 

Reviewer #1: Yes

Reviewer #2: No

4. Have the authors made all data underlying the findings in their manuscript fully available?

Reviewer #1: Yes

Reviewer #2: Yes

5. Is the manuscript presented in an intelligible fashion and written in standard English?

Reviewer #1: Yes

Reviewer #2: Yes

6. Review Comments to the Author

Reviewer #1: (No Response)

Reviewer #2: The authors have addressed most of my earlier concerns and the paper is much improved. The one major issue I have is with the statistics. In some cases, the authors are correlating the proportion of one type of call with the proportion of another (i.e., percentage of trills against the percentage of flat calls) (e.g., Fig 5 E & F, Fig. 9, Fig 10 O and P). As I explained in my earlier comments, this will tend to over-estimate the relationship between calls in a negative direction. To reiterate, if you have just two categories of calls then the correlation between the proportion of calls will, trivially, be perfectly anticorrelated. If you have three categories of calls, then higher proportions of one call do not guarantee a lower proportion of either of the other two call types, but, on average, there will still be a strong tendency to find a negative correlation. In other words, the correlation of proportions is a *biased estimator*. Hence, we cannot know for certain whether the -.75 correlation between flat and trill calls is due to mathematical necessity or a true tendency of rats that emit lots of trills to also reduce the number of flats. The authors have argued in essence (lines 532-538) that, because not all of their correlations are negative, we can trust the correlations that are. However, with three or more proportions adding to 1, the mathematics doesn’t guarantee that all correlations will be negative. It merely dictates that, on average, one is more likely to find negative correlations. Even if the data is completely random and therefore meaningless, one would still expect to find far more significant negative correlations than one should expect using an unbiased statistic. The use of a biased estimator is invalid and is not appropriate in a published, peer-reviewed paper. My suggestion is to perform these sub-type correlations on the absolute call numbers. If that wipes out all significant effects, then the author can just exclude these results from the manuscript. In my opinion, there are still enough solid results to justify publication.

A similar argument can be made about the correlation of absolute call numbers with total number of calls (e.g., Table 1 and Fig S8). It should not be a surprise that A is correlated with A+B+C+D. Notice how many of the p values in the “Absolute prevalence” column are significant? Again, this is a biased estimator and is invalid. The authors should just remove these estimates. The other part of Table 1 that looks at the percental prevalence to call rate is valid and can be retained.

These statistical concerns are critical and should be addressed before publication. Besides this, I have just a few minor concerns, as listed below.

Lesser Concerns:

Fig 4 (and elsewhere): Referents of call type abbreviations (CX, UR, etc) need to be provided

Fig 7: The results say that one rat emitted more inverted U calls than any other call type. I cannot see this because the inverted U type is apparently missing from the graph (I assume its abbreviation is IU). I see the justification for leaving out certain low prevalence calls, but, in order to illustrate what the authors are claiming, the IU bar should be retained.

Line 574. As I said in my previous comments, one possible explanation for the lack of correlation between SP and any of your vocalization measures could be that your range of SP values is too limited. If all your SP values were identical, then obviously, you’d never find a correlation with any measure. Similarly, when your range of values is small, your ability to detect correlations with other measures is reduced. Other studies, which include an independent variable (i.e., drug vs control) might have a larger range of SP values and hence be better able to detect a correlation between SP and vocalizations. The authors have added the statement “there was substantial inter-rat variability in both SP and USV measures, advantageous for correlational analysis.” I don’t feel that this goes far enough, as this issue goes to the heart of the paper. I suggest that the authors compare the coefficient of variation in their study with that in other studies of SP (to the best of their ability). Cite some ranges from other studies for comparison. If the range in your data is comparable to other studies that include an experimental manipulation, then you can largely put this concern to rest. If other studies have a much larger range, however, then the authors need to include a major caveat in interpreting their null result. I do see that the authors offer two other reasons to trust their null result: the low Spearman’s rho values and the near-significant correlation with sucrose consumption. If the range in the data is low, this would also cause low Spearman’s rho values. Further, given that sucrose preference and total sucrose consumption are measuring different aspects of the same task, I would expect these values to be strongly related, so their near-significant correlation doesn’t convince me that power is sufficient to find weaker effects.

Line 487: “rats were tested in the home-cage environment for 4 days”. I questioned this statement in my previous comments and the authors have responded “We feel our statement is justified.” I am not looking for justification. I simply don’t know what data you are referring to. Looking at Figure 1, I see habituation days and testing days, but no reference to home-cage testing. Similarly, in the methods, there is no mention of home-cage testing. Were there unreported recordings done in the home cage, or does “home-cage” refer to some of the testing days in Figure 1. This need to be clarified.

7. PLOS authors have the option to publish the peer review history of their article (what does this mean?). If published, this will include your full peer review and any attached files.

Reviewer #1: **Yes: **Logan J. Bigelow

Reviewer #2: **Yes: **David R. Euston

---

## [Author Response · Author response to Decision Letter 1]

24 Sep 2022

We would again very much like to thank the reviewer for his comments on the manuscript. As before, our responses are indicated by "AUTH" in the text below (line numbers refer to the revised manuscript). We are in agreement with all comments and believe we have now addressed them satisfactorily. 

In addition to the above, we have made a few small edits to improve clarity and we have modified some figures. In terms of figures, we previously displayed only our 14 defined 50-kHz call subtypes, whereas all the call prevalences was calculated using 16 call categories (i.e. the 14 defined subtypes plus "unclear" and "miscellaneous" categories). We have therefore added these two last call categories to the relevant figures (i.e. Fig 7, Fig 10, S5 Fig and S8 Fig). In addition, in order to better reflect our new analysis of inter-call category correlations, we have shifted data previously shown in Fig 10 (panels O and P) to become new panels in Fig 5.

Review Comments to the Author

Reviewer #1: (No Response)

Reviewer #2: The authors have addressed most of my earlier concerns and the paper is much improved. The one major issue I have is with the statistics. In some cases, the authors are correlating the proportion of one type of call with the proportion of another (i.e., percentage of trills against the percentage of flat calls) (e.g., Fig 5 E & F, Fig. 9, Fig 10 O and P). As I explained in my earlier comments, this will tend to over-estimate the relationship between calls in a negative direction. To reiterate, if you have just two categories of calls then the correlation between the proportion of calls will, trivially, be perfectly anticorrelated. If you have three categories of calls, then higher proportions of one call do not guarantee a lower proportion of either of the other two call types, but, on average, there will still be a strong tendency to find a negative correlation. In other words, the correlation of proportions is a *biased estimator*. Hence, we cannot know for certain whether the -.75 correlation between flat and trill calls is due to mathematical necessity or a true tendency of rats that emit lots of trills to also reduce the number of flats. The authors have argued in essence (lines 532-538) that, because not all of their correlations are negative, we can trust the correlations that are. However, with three or more proportions adding to 1, the mathematics doesn’t guarantee that all correlations will be negative. It merely dictates that, on average, one is more likely to find negative correlations. Even if the data is completely random and therefore meaningless, one would still expect to find far more significant negative correlations than one should expect using an unbiased statistic. The use of a biased estimator is invalid and is not appropriate in a published, peer-reviewed paper. My suggestion is to perform these sub-type correlations on the absolute call numbers. If that wipes out all significant effects, then the author can just exclude these results from the manuscript. In my opinion, there are still enough solid results to justify publication.

AUTH: Thank you for pushing us to think harder about this issue of correlations between different 50-kHz call subtypes. As a result, we have expanded our analysis, using two separate approaches in order to establish whether our observed Spearman rho values can readily be explained by the mathematical tendency for percentage measures to be anti-correlated (please see below). 

These new analyses are presented in Results (Comparisons between all 50-kHz call subtypes, lines 401- 456), and covered in the Discussion (Methodological considerations, lines 513- 548; Inter-call subtype correlations, lines 617-642). 

Most importantly, these analyses have established that the negative correlation that we had reported between the percentage prevalence of flat vs. trill calls remains statistically significant even after controlling for the mathematical anti-correlation issue. 

The first of these two analytical approaches was based on collapsing our data into three call categories (i.e. flat, trill, and non-flat/non-trill), and then running simulations of 24-rat experiments based on drawing random numbers from populations having the same group mean and SD percentage prevalence values as our observed call subtype data. In our second approach, we asked whether a high prevalence of flat or trill calls was associated with a disproportionate shift in the prevalence of any particular non-flat/non-trill call subtypes. To this end, we would have liked to apply the same simulation-based approach to our data set comprising the full range of 50-kHz call categories. However, this proved impossible as the less prevalent call subtypes were not normally distributed. Therefore, in our second approach, we instead determined Spearman correlation (rs) coefficients relating flat or trill call prevalence (expressed, as before, as a percentage of all 50-kHz calls) to the prevalence of every other 50-kHz call subtype (now expressed as a percentage of all non-flat/non-trill 50-kHz calls). 

Please note that this second approach, like the first approach, is non-biased, i.e. it avoids the mathematical anti-correlation issue that you raised. This second approach identified statistically significant correlations between certain call subtypes and either flat or trill calls (the results are shown in Table 1 under "Approach 2").

As per your suggestion, we also use the absolute call prevalence data in order to calculate correlation coefficients between pairs of 50-kHz call subtypes. This analysis produced positive Spearman rho values (mean rho ± SD = + 0.63 ± 0.18 in Phase 1, for example). However, we consider this type of correlation of only limited value, as such high correlations could easily be driven by differences in the overall 50-kHz call rate. This can be seen as follows, using an extreme example. Suppose that our 24 rats differed in their 50-kHz call rate, but that each rat made exactly 40% trill and 30% flat calls. Now imagine a scatterplot showing the trill call rate vs. the flat call rate, with each rat represented by single dot. In such a scenario, there would be a perfect positive correlation between the absolute prevalence (e.g. calls/min) of flat calls and trill calls.

A similar argument can be made about the correlation of absolute call numbers with total number of calls (e.g., Table 1 and Fig S8). It should not be a surprise that A is correlated with A+B+C+D. Notice how many of the p values in the “Absolute prevalence” column are significant? Again, this is a biased estimator and is invalid. The authors should just remove these estimates. The other part of Table 1 that looks at the percental prevalence to call rate is valid and can be retained.

AUTH: Apologies for overlooking this point. To avoid this bias, we have now correlated the absolute prevalence (calls/min) of each subtype with the 50-kHz call rate generated from summing all the remaining subtypes (e.g., correlating A with B+C+D+E etc). In this new analysis, the absolute prevalences of almost all subtypes are still positively correlated with the call rate. While conducting this analysis, we also found that we had not previously accounted for the fact that some rats were time-sampled. We have now corrected this oversight (which pertains only to this analysis), and as a result the correlation coefficients are actually somewhat higher than before, strengthening our conclusion. Please refer to lines 480-486 of the Results and lines 595-605 of the Discussion.

These statistical concerns are critical and should be addressed before publication. Besides this, I have just a few minor concerns, as listed below.

LESSER CONCERNS (Reviewer 2)

Fig 4 (and elsewhere): Referents of call type abbreviations (CX, UR, etc) need to be provided

AUTH: Referents have been added for Fig 4, (lines 301-305) and Fig 7 (lines 362- 365).

Fig 7: The results say that one rat emitted more inverted U calls than any other call type. I cannot see this because the inverted U type is apparently missing from the graph (I assume its abbreviation is IU). I see the justification for leaving out certain low prevalence calls, but, in order to illustrate what the authors are claiming, the IU bar should be retained.

AUTH: We did not identify any of the subtypes as missing from the figures, including IU/inverted-U (Fig 7 and S5 Fig). However, we have clarified this result and referred to the relevant panels in the manuscript (lines 356-358). Overall, two rats preferentially emitted a subtype other than flat or trill calls; short calls in one case (Rat 13 in Fig 7 and Rat 9 in S5 Fig) and inverted-U calls in the other (Rat 21, phase 2, shown in S5 Fig). 

Line 574. As I said in my previous comments, one possible explanation for the lack of correlation between SP and any of your vocalization measures could be that your range of SP values is too limited. If all your SP values were identical, then obviously, you’d never find a correlation with any measure. Similarly, when your range of values is small, your ability to detect correlations with other measures is reduced. Other studies, which include an independent variable (i.e., drug vs control) might have a larger range of SP values and hence be better able to detect a correlation between SP and vocalizations. The authors have added the statement “there was substantial inter-rat variability in both SP and USV measures, advantageous for correlational analysis.” I don’t feel that this goes far enough, as this issue goes to the heart of the paper. I suggest that the authors compare the coefficient of variation in their study with that in other studies of SP (to the best of their ability). Cite some ranges from other studies for comparison. If the range in your data is comparable to other studies that include an experimental manipulation, then you can largely put this concern to rest. If other studies have a much larger range, however, then the authors need to include a major caveat in interpreting their null result. I do see that the authors offer two other reasons to trust their null result: the low Spearman’s rho values and the near-significant correlation with sucrose consumption. If the range in the data is low, this would also cause low Spearman’s rho values. Further, given that sucrose preference and total sucrose consumption are measuring different aspects of the same task, I would expect these values to be strongly related, so their near-significant correlation doesn’t convince me that power is sufficient to find weaker effects.

AUTH: In order to compare the CV values from our study to the literature, we initially used Clarivate Web of Science to search for the most cited papers under the following search term “sucrose preference” AND “rat.” Nine studies were selected based on the following criteria: tested rats (i.e., not mice); avoided sucrose concentrations higher than 1%; and mean, SD or SEM, and n values were stated in the text or could be estimated from a figure. In the same way, 14 additional studies were selected from a meta-analysis on the chronic mild stress procedure by Antoniuk et al. (2019), doi: https://doi.org/10.1016/j.neubiorev.2018.12.002.

In our own study, we report a coefficient of variation (CV) of 0.10 for our sucrose preference data (line 381). Published studies generally yielded higher CV values (control group mean CV = 0.15 and stressed group mean CV = 0.26; see new S3 Table and reference list below). However, most of these published studies reported sucrose preference data from a single session, or at best a few test sessions, such that it is unclear how much of the observed variability represented random noise vs. consistent individual differences. In our study, in contrast, the sucrose preference test was conducted numerous times on each animal, so that our sucrose preference values are likely to be more reliable, and our CV value less reflective of random noise. These data have been included as a supplementary table (S3 Table) and covered in the Discussion section of the manuscript (lines 553-560, and lines 664-668).

Line 487: “rats were tested in the home-cage environment for 4 days”. I questioned this statement in my previous comments and the authors have responded “We feel our statement is justified.” I am not looking for justification. I simply don’t know what data you are referring to. Looking at Figure 1, I see habituation days and testing days, but no reference to home-cage testing. Similarly, in the methods, there is no mention of home-cage testing. Were there unreported recordings done in the home cage, or does “home-cage” refer to some of the testing days in Figure 1. This need to be clarified.

AUTH: We agree that this wording was confusing. We did not test the rats in a home-cage environment at any point in our study. As we now make clear, we were actually referring to the Schwarting et al. 2007 study mentioned in the immediately preceding sentence (lines 569-570). More specifically, Schwarting et al. 2007 recorded a group of rats in their home cages (albeit in a separate testing room). The data for home cage recorded USV emission can be seen in Figures 2-4 in their paper (https://doi.org/10.1016/j.bbr.2007.01.029). This was the only paper we found that had measured emission of USVs without an eliciting stimulus (ex: drugs or play), and was therefore comparable to our study which also measured USVs in a non-evoked state.

---

## [Decision Letter · Decision Letter 2]

13 Oct 2022

Stable long-term individual differences in 50-kHz vocalization rate and call subtype prevalence in adult male rats: comparisons with sucrose preference

PONE-D-22-12721R2

Dear Dr. Clarke,

We’re pleased to inform you that your manuscript has been judged scientifically suitable for publication and will be formally accepted for publication once it meets all outstanding technical requirements.

Kind regards,

Brenton G. Cooper, Ph.D.

Academic Editor

PLOS ONE

Additional Editor Comments (optional):

Reviewers' comments:

Reviewer's Responses to Questions

**Comments to the Author**

1. If the authors have adequately addressed your comments raised in a previous round of review and you feel that this manuscript is now acceptable for publication, you may indicate that here to bypass the “Comments to the Author” section, enter your conflict of interest statement in the “Confidential to Editor” section, and submit your "Accept" recommendation.

Reviewer #2: All comments have been addressed

2. Is the manuscript technically sound, and do the data support the conclusions?

Reviewer #2: Yes

3. Has the statistical analysis been performed appropriately and rigorously? 

Reviewer #2: Yes

4. Have the authors made all data underlying the findings in their manuscript fully available?

Reviewer #2: Yes

5. Is the manuscript presented in an intelligible fashion and written in standard English?

Reviewer #2: Yes

6. Review Comments to the Author

Reviewer #2: The authors have addressed my concerns about bias in their statistics. I agree that the correlations between absolute call prevalence are of limited value and the authors might want to just exclude them. However, to address one of their comments, it was not a forgone conclusion that all such correlations would be high. Referring to their example where all rats make 40% trill and 30% flat, this example presupposes that call rates for individual calls are proportional to overall call rate. However, it could have been the case that, as overall call rate increases, rats keep a fixed rate of flat calls and all increases are due to added trills. Hence, the proportion of trills would increase with call rate and there would be no relationship between flat and trill calls. The fact that the rising tide of overall call rate raises all boats evenly is interesting. I think we all suspected this, but the author’s results demonstrate this nicely.

All my other concerns have been addressed. To the authors: nice work.

7. PLOS authors have the option to publish the peer review history of their article (what does this mean?). If published, this will include your full peer review and any attached files.

Reviewer #2: **Yes: **David R Euston

---

## [Editor Report · Acceptance letter]

20 Oct 2022

PONE-D-22-12721R2 

Stable long-term individual differences in 50-kHz vocalization rate and call subtype prevalence in adult male rats: comparisons with sucrose preference 

Dear Dr. Clarke:

I'm pleased to inform you that your manuscript has been deemed suitable for publication in PLOS ONE. Congratulations! Your manuscript is now with our production department. 

Kind regards, 

on behalf of

Dr. Brenton G. Cooper 

Academic Editor

PLOS ONE